# Pluralistic Leaderboards

**Nika Haghtalab** [1]  **Ariel D. Procaccia** [2]  **Han Shao** [3]  **Serena Lutong Wang** [2]  **Kunhe Yang** [1]

## Abstract

Recent leaderboard-based evaluations of large language models aggregate user feedback by fitting a Bradley–Terry model to pairwise comparisons, producing a single global ranking based on a latent quality score. While appealing for its simplicity, this approach is incompatible with heterogeneous preferences: when LLMs are used across diverse tasks and use cases, users who favor fundamentally different model behaviors can be systematically misrepresented when collapsed into a single quality score. To address this issue, we study *pluralistic leaderboards* that aim to remain *stable* with respect to heterogeneous user populations. Drawing on ideas from social choice theory, we adapt the notion of *local stability*, which requires that no model outside the top-$k$ positions is collectively preferred to the top-$k$ set by more than $O(1/k)$ fraction of users. Building on techniques from the social choice literature, we design an alternative leaderboard mechanism that satisfies local stability while eliciting only $\widetilde{O}(k)$ pairwise comparisons per user, where $k$ is the size of the prefix for which stability is guaranteed. Using data from LMArena, we show that standard Bradley–Terry aggregation can violate local stability in practice, whereas our method provides substantially stronger stability guarantees.

## 1. Introduction

Until recently, leading approaches to the evaluation of large language models (LLMs) relied on benchmarks, which measure performance on multiple tasks and aggregate the corresponding scores (Liang et al., 2023; Jimenez et al., 2024). While this paradigm has provided a standardized and reproducible basis for comparing models, it is increasingly strained by rapid model iteration, benchmark saturation, and sensitivity to task selection; it also struggles to capture qualitative dimensions of model behavior such as helpfulness.

Some of these shortcomings are addressed by an alternative evaluation approach championed by LMArena (Chiang et al., 2024). Upon submitting a prompt, an LMArena user is shown a pair of responses from two different LLMs and is asked to indicate which is preferred. LMArena aggregates these votes by fitting a Bradley–Terry (BT) model. That is, it posits that each LLM has a single latent "quality" score, where the probability that one model beats another on a prompt depends only on the difference between their qualities, independently of the identity of the user. By computing the maximum likelihood estimator for the quality of each LLM based on the input comparisons, the platform is able to produce a ranking over the models.

The approach taken by LMArena is compelling and hugely successful,[1] but it has also attracted notable controversy and criticism (Singh et al., 2025). In our view, its main conceptual flaw is its reliance on the BT model, and with it, the implicit assumption that all pairwise comparisons originate from a generic (or "average") person — a view that is fundamentally at odds with heterogeneous preferences. People come to LLMs with different goals (e.g., safety vs. creativity, conciseness vs. thoroughness, formality vs. warmth) that represent inherently different latent utility models. These differences can be large enough that there may be no single ground-truth scalar "quality" that explains society's preferences. Statistically speaking, a single BT model is a misspecified model of the exhibited comparisons, and as such, the maximum-likelihood BT scores need not represent what people want in any meaningful way (Ge et al., 2024; Gölz et al., 2025; Shirali et al., 2025).

When fundamental disagreements arising from heterogeneous preferences are unavoidable, social choice theory (Brandt et al., 2016) provides a principled framework for aggregation. To see how these issues manifest in LMArena, and how social choice provides solutions, consider a setting with six competing LLMs $\{a, b, c, d, e, f\}$, and two

[1]University of California, Berkeley [2]Harvard University [3]University of Maryland, College Park. Correspondence to: Kunhe Yang <kunheyang@berkeley.edu>.

*Proceedings of the 43rd International Conference on Machine Learning*, Seoul, South Korea. PMLR 306, 2026. Copyright 2026 by the author(s).

---

[1]As of January 2026, LMArena is valued at $1.7B just months after it launched as a company.

preference profiles shown below.

| 60% | 40% |
|-----|-----|
| $a$ | $f$ |
| $b$ | $e$ |
| $c$ | $d$ |
| $d$ | $c$ |
| $e$ | $b$ |
| $f$ | $a$ |

| 10% | 10% | 10% | 10% | 10% | 10% | 10% | 10% | 10% | 10% |
|-----|-----|-----|-----|-----|-----|-----|-----|-----|-----|
| $a$ | $c$ | $c$ | $a$ | $b$ | $a$ | $a$ | $b$ | $b$ | $a$ |
| $b$ | $b$ | $d$ | $e$ | $e$ | $d$ | $d$ | $c$ | $c$ | $c$ |
| $c$ | $d$ | $a$ | $b$ | $d$ | $e$ | $e$ | $e$ | $e$ | $d$ |
| $f$ | $f$ | $f$ | $f$ | $f$ | $f$ | $f$ | $f$ | $f$ | $f$ |
| $d$ | $e$ | $b$ | $d$ | $a$ | $c$ | $b$ | $a$ | $d$ | $e$ |
| $e$ | $a$ | $e$ | $c$ | $c$ | $b$ | $c$ | $d$ | $a$ | $b$ |

*Profile 1*          *Profile 2*

The two profiles induce the same pairwise comparisons and are therefore indistinguishable from the perspective of the BT model, which outputs the ranking $a \succ b \succ c \succ d \succ e \succ f$ in both cases. Now suppose (for simplicity) that a user tests the leaderboard's top three models to find one that is to their liking. For Profile 2, the set $\{a, b, c\}$ works well, as the favorite model of each and every individual is included. By contrast, the same outcome is far less satisfactory for Profile 1. Here, 40% of the population is only exposed to models that they dislike; an outcome that elevates $f$ into the top tier would ensure that this sizable faction is not shut out.

Social choice theory provides a formal understanding of the principle that a large and cohesive faction should be fairly represented in the outcome. There is a significant literature on representation in committee elections — building on the work of Aziz et al. (2017a) — which has led to deployed participatory budgeting methods (Peters et al., 2021) and AI-driven applications (Fish et al., 2024). As we discuss below, however, it is nontrivial to adapt these techniques to LLM leaderboards, where the input consists of (typically sparse) pairwise comparisons. Our research problem, therefore, is this:

> Design *pluralistic leaderboards* that take pairwise comparisons as input and produce an outcome that fairly represents heterogeneous preferences.

**Our approach and results.** To construct pluralistic leaderboards, we need a formal notion for *fair representation*; that is, any large cohesive subgroups of users should have sufficient impact on the leaderboard to ensure that the models they collectively favor are ranked near the top. To this end, we adopt the notion of *local stability* (Aziz et al., 2017b) from the social choice literature on committee elections. Intuitively, local stability requires that no sufficiently large subgroup of users can "profitably deviate" to an alternative candidate that is excluded from the selected committee. More formally, a committee $W_k$ of size $k$ is *locally stable* if for any candidate $a \notin W_k$, the fraction of users who prefer $a$ to all candidates in $W_k$ is at most $1/k$, and is $\gamma$-approximately local stable if the fraction is at most $\gamma/k$.

Since our ultimate goal is to produce *rankings* rather than committees, we extend local stability to rankings by requiring the property to hold for every prefix of the ranking.

Specifically, we treat the top-$k$ prefix of a ranking as a size $k$ committee, and impose local stability at every cutoff $k$. Our goal is therefore to design a ranking mechanism that converts pairwise comparisons into such an (approximately) locally stable ranking.

We begin with studying the committee selection setting, where we build on algorithms by Jiang et al. (2020). The challenge is that these algorithms assume access to the complete rankings of all the users. We adapt these techniques to the leaderboard setting where users are drawn from an underlying distribution, and they each contribute only a small amount of feedback, rather than providing full rankings. By carefully controlling the errors arising from finite-sample estimation, we develop an algorithm that produces approximately locally stable committees, with a constant approximation factor. Our algorithm uses $\widetilde{O}(\text{poly}(m))$ sampled users and $\widetilde{O}(k)$ pairwise comparison queries per user, where $k$ denotes the committee size. We then develop a reduction that uses this algorithm as a subroutine to construct approximately stable rankings, incurring only a constant multiplicative factor in the approximation ratio.

Finally, we empirically validate our approach on synthetic and semi-synthetic experiments derived from LMArena data. Our experiments demonstrate that the currently-used Bradley–Terry ranking can violate stability, whereas our methods achieve stability to a degree that is even stronger than our theoretical bounds in practice.

**Related Works.** While a global leaderboard aggregates all battles into a single score, recent work has attempted to incorporate heterogeneity by constructing task-specific or more fine-grained leaderboards. For example, Frick et al. (2025) propose Prompt-to-Leaderboard, which conditions Bradley–Terry scores on the prompt to produce prompt-dependent rankings from LMArena data. We take a different approach: rather than producing multiple task- or prompt-specific rankings, we aim to construct a single global ranking that automatically represents all sufficiently large user subgroups, whether defined explicitly or implicitly.

Our approach builds on the social choice literature on local stability in committee selection. In particular, Jiang et al. (2020) propose algorithms for computing approximately locally stable committees, and subsequent work improves the approximation factors (Charikar et al., 2025; Nguyen et al., 2025). These results assume access to richer preference information (users' full rankings) than is typically available in leaderboard settings. We extend this line of work by adapting local stability algorithms to sparse pairwise comparisons and by lifting committee-level guarantees to full rankings.

Finally, our work is related to research on ranking and committee selection under incomplete or partially elicited preferences (Halpern et al., 2023; 2024; Springham et al., 2025). In particular, Halpern et al. (2023) and Springham et al.

(2025) study representation under approval-based preferences, whereas our framework assumes ranked preferences revealed through pairwise comparisons.

## 2. Model and Preliminaries

We use $\mathcal{C} = \{1, 2, \ldots, m\}$ to denote the set of $m$ models competing on the leaderboard. We assume that users are heterogeneous and are drawn from a distribution $\mathcal{D}$. Each user $i$ is associated with a strict preference order over the models in $\mathcal{C}$, represented by a permutation $\sigma_i = (\sigma_i(1), \ldots, \sigma_i(m)) \in \mathsf{Sym}(\mathcal{C})$, where $\sigma_i(1) \succ_i \sigma_i(2) \succ_i \ldots \succ_i \sigma_i(m)$. Here, $\mathsf{Sym}(\mathcal{C})$ denotes the set of all permutations of $\mathcal{C}$ and $\mathcal{D}$ can be equivalently viewed as a distribution over rankings.

**Leaderboard Mechanism.** The goal of the leaderboard is to produce a ranking of models $\pi = (\pi(1), \ldots, \pi(m)) \in \mathsf{Sym}(\mathcal{C})$ based on the preferences of the users. Users interact with the leaderboard by contributing "battles", which are pairwise comparisons between two models. Specifically, for each battle, a user $i$ submits a prompt to the leaderboard platform, and the platform selects a pair of models $\{x, y\} \subseteq \mathcal{C}$ and presents their outputs to the user without revealing the identity of the models. The user then indicates which output they prefer.

If user $i$ participates in $d$ battles, we denote the outcomes by a sequence of ordered pairs $(x_i^t \succ_i y_i^t)_{t=1}^d$. We make the key simplifying assumption that users are consistent: for any user $i$ and any pair of models $x, y \in \mathcal{C}$, $i$ prefers $x$ over $y$ in a battle if and only if $x$ appears before $y$ in the associated ranking $\sigma_i$.[2]

A leaderboard's *ranking mechanism* $\mathcal{M}$ specifies both which models are selected for comparison and how the observed battles are aggregated into a ranking. Specifically, $\mathcal{M}$ has sampling access to the population $\mathcal{D}$, and can query each sampled user for their preferences via a sequence of pairwise comparison queries. The choice of which model pairs to compare may be adaptive and depend on the outcomes of previous queries. We measure the complexity of $\mathcal{M}$ by (i) the maximum number of pairwise comparisons elicited per user, denoted with $d$, and (ii) the total number of users sampled from $\mathcal{D}$, denoted with $n$.

**Bradley–Terry Ranking by LM Arena (Chiang et al., 2024).** We describe the Bradley–Terry-based leaderboard mechanism proposed by Chiang et al. (2024) for LMArena, which we denote by $\mathcal{M}_{\mathsf{BT}}$. This mechanism first uses an active sampling approach for selecting model pairs, then ranks models according to scores obtained by fitting a Bradley–

Terry (BT) model to the observed battle outcomes.

In this approach, the selection of sampled pairs is independent of the user's identity: for the $t$-th battle in the data collection process, the leaderboard samples an unordered pair of models $\{x^t, y^t\} \sim P_t$ where $P_t \in \Delta(\mathcal{C}^2)$ is a distribution over model pairs. The distribution $P_t$ is chosen adaptively to prioritize pairs that are expected to reduce the uncertainty in the estimated BT scores, and is shown by Chiang et al. (2024) to converge almost surely to a limiting distribution as $t \to \infty$.

Given a total of $T$ battle outcomes $(x^t \succ y^t)_{t=1}^T$, Chiang et al. (2024) estimate BT scores by solving the following reweighted maximum likelihood estimation problem:

$$\theta = \underset{\theta \in \mathbb{R}^m}{\mathrm{argmax}} \sum_{t=1}^T \frac{1}{P(\{x^t, y^t\})} \log \frac{1}{1 + e^{-(\theta_{x^t} - \theta_{y^t})}},$$

where $\theta_x$ is the BT score of model $x$. The final leaderboard ranking orders models according to their BT scores $\theta_x$.[3]

**Local Stability.** In this paper, we consider the local stability of the leaderboard. Local stability, proposed by Aziz et al. (2017b), is a property of a committee that ensures that no outside model is collectively preferred by a sufficiently large subgroup of users. Formally, a committee $W_k \subseteq \mathcal{C}$ of size $k$ is said to be $\gamma$-*approximately locally stable* if

$$\max_{a \notin W_k} \Pr_{i \sim \mathcal{D}}[a \succ_i W_k] \le \frac{\gamma}{k},$$

where $a \succ_i W_k$ means that user $i$ prefers $a$ to all models in $W_k$. When $\gamma = 1$, the committee is said to be locally stable. We refer to the smallest $\gamma$ that satisfies the above condition as the *stability ratio* of $W_k$.

Since our ultimate goal is to produce a full ranking of models, we extend this notion to rankings by considering the local stability of all *prefix committees*. For a ranking $\pi \in \mathsf{Sym}(\mathcal{C})$, let $W_k^\pi = \{\pi(1), \ldots, \pi(k)\}$ denote the prefix committee consisting of the top $k$ models in the ranking $\pi$. We say that $\pi$ is $\gamma$-*approximately locally stable ranking* for user distribution $\mathcal{D}$ if, for every $k \le m$, the prefix committee $W_k^\pi$ is $\gamma$-approximately locally stable.

## 3. Committee Selection

We consider the committee selection problem as a first step towards constructing stable rankings. Our goal is to select a single committee of a given size that satisfies approximate stability. We will later show in Section 4 how this proce-

---

[2]In practice, a user's preferences may also depend on contextual factors such as the task or use case. Our model can capture this by treating the user's type as a latent type that aggregates all such factors, so that preferences are fixed conditioned on the user's type.

[3]The current mechanism by LMArena extends this by calculating the confidence intervals for each model's estimated BT score, and displaying ties in the ranking when intervals overlap (LMArena Team, 2025b).

---

**Algorithm 1** Committee Selection via Iterated Rounding

---

**Input:** Committee size $k$; sampling oracle $\mathcal{D}$, rank estimation oracle $\widehat{\mathsf{Rank}}$ (Algorithm 5)

**Output:** A committee $\hat{S}$ of size $k$.

**Parameters:** $\alpha \leftarrow \frac{1}{2} + 4\varepsilon$, $\beta \leftarrow \frac{1}{4} + 2\varepsilon$. Number of Rounds $T \leftarrow \lceil 10 \log(\frac{k}{\varepsilon}) \rceil$, sub-committee sizes $(k_t)_{t \in [T]}$ defined by $k_t \leftarrow \max\{1, \lfloor (1-\alpha)\alpha^{t-1}k \rfloor\}$ while the total committee size is less than $k$, and stop once $\sum_t k_t = k$.

**Initialize** $\hat{S} \leftarrow \emptyset$

**for** $t = 1$ **to** $T$ **do**

$\widehat{\mathcal{D}}_t^{\beta-\varepsilon} \leftarrow$ sampling oracle for unsatisfied users at round $t$, implemented in Algorithm 4 with history $(S_\tau, \Delta_\tau)_{\tau \leq t-1}$ and threshold $\beta - \varepsilon$;

$\Delta_t \leftarrow (1+\varepsilon)$-approximately stable lottery for $\widehat{\mathcal{D}}_t^{\beta-\varepsilon}$ of size $k_t$ (see Cheng et al. (2020));

$S_t \leftarrow$ committee in $\mathrm{supp}(\Delta_t)$ that minimizes estimated unsatisfied probability $\frac{1}{N} \sum_{i=1}^{N} \mathbb{1}\left[ \widehat{\mathsf{Rank}}(i; S, \Delta) \leq \beta - \varepsilon \right]$, for $N$ i.i.d. users drawn from $\widehat{\mathcal{D}}_t^{\beta-\varepsilon}$;

$\hat{S} \leftarrow \hat{S} \cup S_t$

**end for**

**Return** $\hat{S}$;

---

dure can be adapted or used as a subroutine to construct approximately stable rankings.

Our committee selection algorithm (Algorithm 1) is closely inspired by the *Iterated Rounding* Algorithm proposed by Jiang et al. (2020). But in contrast to their setting, which assumes full access to the users' complete rankings, we adapt the approach to the sampling-based model in which the user population is represented by an underlying distribution that can be accessed only through sampling, and each user's preferences can be accessed through only a limited number of pairwise comparisons.

At a high level, the Iterated Rounding approach proceeds in $T = \tilde{O}(\log k)$ rounds. In each round $t$, it identifies a sub-committee $S_t$ of geometrically decreasing size. This sub-committee is chosen to provide a good representation for a large fraction of the users that have not been well represented by sub-committees selected in previous rounds. Repeating this process guarantees that the algorithm *makes progress* by geometrically shrinking the fraction of poorly represented users, and as a result, the union of all such sub-committees forms a committee of size $k$ with the desired stability guarantees.

Specifically, we formalize what we mean by *representation* in the above discussion, which depends on how each sub-committee is selected in *Iterated Rounding*. In each round $t$, the algorithm first computes a $\gamma$-approximately stable lottery $\Delta_t$ — a distribution over size-$k_t$ committees — for

the currently unsatisfied user distribution. It then selects a committee $S_t$ from the support of $\Delta_t$.

To quantify representation, we define the *rank* of a committee $S$ for user $i$ under lottery $\Delta$ as

$$\mathsf{Rank}(i; S, \Delta) := \Pr_{S' \sim \Delta}[S \succeq_i S'],$$

where $S \succeq_i S'$ if user $i$'s favorite candidate in $S$ is the same or more preferred than their favorite candidate in $S'$. Intuitively, a larger rank indicates that $S$ is more representative of $\Delta$ for user $i$. Given a threshold $x \in (0, 1)$, we say that user $i$ is *satisfied* at the $t$-th round if $\mathsf{Rank}(i; S_t, \Delta_t) \geq x$. Following this, the users that remain *unsatisfied* at round $t$ are

$$U_t^x = \{i \mid \mathsf{Rank}(i; S_\tau, \Delta_\tau) < x, \ \forall \tau < t\}.$$

Accordingly, we use $\mathcal{D}_t^x$ to denote the conditional distribution of unsatisfied users, which is $\mathcal{D}_t^x := \mathcal{D} \mid U_t^x$.

Algorithm 1 is obtained by adapting the above iterated rounding framework to a sampling-based and incomplete-preference setting. In particular, the estimation of user ranks, selection of sub-committees in each round, and the identification of unsatisfied users must all be carried out using only finite samples and pairwise comparison queries, rather than exact access to users' complete rankings.

In the remainder of this section, we formally establish the guarantees achieved by Algorithm 1, showing how the algorithm maintains approximate local stability despite the errors introduced by finite-sample estimation.

**Theorem 3.1** (Guarantee of Algorithm 1)**.** *For any user distribution $\mathcal{D}$ and committee size $k$, Algorithm 1 returns a committee $\hat{S}$ of size at most $k$, such that with probability at least $1 - \delta$, $\hat{S}$ is $(16 + O(\varepsilon))$-approximately stable for $\mathcal{D}$. The algorithm has the following complexity:*

*(i) The total number of users sampled is $n = O(\mathrm{poly}(m, 1/\varepsilon, \log 1/\delta))$;*

*(ii) The maximum number of pairwise comparisons per user is $d = O(\frac{k}{\varepsilon^2} \log(\frac{m}{\varepsilon\delta}))$.*

*In particular, when $\varepsilon$ is set to a constant, Algorithm 1 achieves a constant-factor approximation using $\tilde{O}(k)$ pairwise comparisons per user.*

Below we provide a proof sketch for a simpler case where $\mathcal{D}$ has $\mathrm{poly}(m, 1/\varepsilon, \log(k), \log(1/\delta))$. The full proof for the general case is relegated to Appendix B.

*Proof sketch of Theorem 3.1.* To show that the final committee $\hat{S}$ is approximately stable, we can decompose the user population into $T$ rounds based on when a user becomes satisfied. Recall that we have used $U_t^x$ to denote the users that remain unsatisfied at round $t$, measured using

their true rank. However, since each user only answers a limited number of queries, the quantity $\mathrm{Rank}(i; S, \Delta)$ cannot be computed directly. Instead, we estimate it by querying user $i$ to compare $S$ against a finite number of committees $S'_1, \ldots, S'_L \overset{\text{i.i.d.}}{\sim} \Delta$, and define

$$\widehat{\mathrm{Rank}}(i; S, \Delta) := \frac{1}{L} \sum_{\ell=1}^{L} \mathbb{1}\left[S \succeq_i S'_\ell\right].$$

Here, each query $\mathbb{1}\left[S \succeq_i S'_\ell\right]$ can be implemented by querying user $i$ with $|S| + |S'|$ pairwise comparisons. By standard concentration arguments, this estimator achieves at most $\varepsilon$ additive error with $\widetilde{O}(1/\varepsilon^2)$ draws from $\Delta$ (see Fact B.5).

With the estimated rank, we use

$$\widehat{U}_t^x := \left\{ i \mid \widehat{\mathrm{Rank}}(i; S_\tau, \Delta_\tau) \leq x, \, \forall \tau < t \right\}$$

to denote the subset of users that remain unsatisfied at round $t$, measured using their estimated rank.

Now we decompose the user population into $T + 1$ subgroups, where the $t$-th subgroup is the users that become satisfied at round $t$, i.e., $U_t^x \setminus U_{t+1}^x$ for $t \leq T$, and the remaining users are $\widehat{U}_{T+1}^x$ for $t = T + 1$. By combining the argument in Jiang et al. (2020) with estimation errors, we can show that the newly satisfied users are well-represented by the committee $S_t$, as shown by the following lemma:

**Lemma 3.2** (Guarantee for satisfied users)**.** *Let $S_t$ be the sub-committee selected at round $t$, and $x$ be the threshold used to measure satisfaction with the estimated rank. We have that for any alternative $a \notin S_t$,*

$$\Pr_{i \sim \widehat{\mathcal{D}}_t^x}\left[a \succ_i S_t \text{ and } i \in \widehat{U}_t^x \setminus \widehat{U}_{t+1}^x\right] \leq \frac{1 + \varepsilon}{k_t \cdot (x - \varepsilon)}.$$

*In other words, users who are satisfied in the $t$-th round (i.e., users in $\widehat{U}_t^x \setminus \widehat{U}_{t+1}^x$) have bounded probability of deviation.*

More importantly, we must ensure that the algorithm continues to make steady progress by shrinking the fraction of unsatisfied users, even when we are using estimated ranks with some additive errors. This crucially depends on our choice of the threshold $x$. We employ a *more conservative threshold* of $x = \beta - \varepsilon$ when updating the distribution of unsatisfied users, as a proxy for the intended threshold $\beta$. This choice is designed to account for estimation error and to ensure that users who are already satisfied are successfully removed from the estimated unsatisfied set, i.e., $\widehat{U}_t^{\beta - \varepsilon} \subseteq U_t^\beta$. We show in the following lemma that, with this conservative threshold, the probability mass of estimated unsatisfied users decreases geometrically across rounds.

**Lemma 3.3** (Geometrically shrinking unsatisfied users)**.** *For all $t \geq 1$, when the threshold is set to be $x = \beta - \varepsilon$, the probability mass of estimated unsatisfied users shrink geometrically: with probability at least $1 - T\delta$,*

$$\widehat{\mu}_t := \Pr_{i \sim \mathcal{D}}[i \in \widehat{U}_t^{\beta - \varepsilon}] \leq (\beta + 2\varepsilon)^{t-1} \quad \text{for all } t \in [T].$$

Putting the two lemmas together, we have the following guarantee:

$$\Pr_{i \sim \mathcal{D}}[a \succ_i \hat{S}]$$

$$\overset{(a)}{\leq} \sum_{t=1}^{T} \Pr_{i \sim \mathcal{D}}\left[a \succ_i \hat{S}, i \in \widehat{U}_t^{\beta - \varepsilon} \setminus \widehat{U}_{t+1}^{\beta - \varepsilon}\right] + \Pr_{i \sim \mathcal{D}}\left[\widehat{U}_{T+1}^{\beta - \varepsilon}\right]$$

$$\overset{(b)}{\leq} \sum_{t=1}^{T} \Pr_{i \sim \mathcal{D}}\left[a \succ_i S_t, i \in \widehat{U}_t^{\beta - \varepsilon} \setminus \widehat{U}_{t+1}^{\beta - \varepsilon}\right] + \Pr_{i \sim \mathcal{D}}\left[\widehat{U}_{T+1}^{\beta - \varepsilon}\right]$$

$$= \sum_{t=1}^{T} \widehat{\mu}_t \cdot \Pr_{i \sim \widehat{\mathcal{D}}_t^{\beta - \varepsilon}}\left[a \succ_i S_t, i \in U_t^{\beta - \varepsilon} \setminus U_{t+1}^{\beta - \varepsilon}\right] + \widehat{\mu}_{T+1}$$

$$\overset{(c)}{\leq} \sum_{t=1}^{T} \frac{(\beta + 2\varepsilon)^{t-1} \cdot (1 + \varepsilon)}{\alpha^{t-1}(1 - \alpha)k \cdot (\beta - 2\varepsilon)} + (\beta + 2\varepsilon)^{T+1}.$$

In the above argument, (a) follows from dividing the user population by the first time they are removed from the estimated unsatisfied set; (b) leverages the fact that $a \succ_i \hat{S}$ implies $a \succ_i S_t$; (c) follows from combining Lemma 3.2 (with $x = \beta - \varepsilon$) and Lemma 3.3.

Finally, we optimize the parameters $\alpha, \beta, \varepsilon$ to minimize the approximation factor. By setting $\alpha = \frac{1}{2} + 4\varepsilon$, $\beta = \frac{1}{4} + 2\varepsilon$, and $T = \lceil 10 \log(k/\varepsilon) \rceil$, the final bound is at most $16 + O(\varepsilon)$. For example, setting $\varepsilon = 0.01$ and $0.05$ gives us approximation factors of 18.97 and 39.2, respectively. $\quad\square$

## 4. From Committees to Rankings

Building from the committee selection algorithm, we next construct algorithms that output full rankings over models. Recall that we extend the *local stability* notion from committee selection to rankings by requiring that, for every $k \leq m$, the top-$k$ prefix of the ranking forms an (approximately) locally stable committee. This prefix-based formulation captures the fair representation property we seek at every cutoff of the leaderboard ranking.

The connection between committee selection and rankings is conceptually natural and has been explored in prior work (Elkind et al., 2017; Skowron et al., 2017; Aziz et al., 2025), especially through *committee monotonicity*, which requires that the committee selected for size $k$ be a subset of the committee selected for size $k + 1$ for all $k$.[4] Under this

---

[4]Aziz et al. (2025) focuses on achieving committee monotonicity and the PSC (Proportionality for Solid Coalition) notion under

---

**Algorithm 2** Ranking via Geometric Checkpoints

---

**Input:** Sampling oracle $\mathcal{D}$; committee selection subroutine COMMITTEE (Algorithm 1)
**Output:** A ranking $\pi \in \mathsf{Sym}(\mathcal{C})$
**Parameter:** Growth factor $\lambda > 1$, $R \leftarrow \lceil \log_\lambda m \rceil$
$\pi \leftarrow [\,]$
**for** $r = 1, \ldots, R$ **do**
  $k_r \leftarrow \lfloor \lambda^{r-1} \rfloor$ (size of the next checkpoint committee)
  $A_r \leftarrow$ COMMITTEE$(k_r, \mathcal{D})$
  Append $A_r \setminus A_{\leq r-1}$ to $\pi$, of arbitrary order or as ties
  (where $A_{\leq r-1} := \bigcup_{s \leq r-1} A_s$)
  **if** $A_{\leq r} = \mathcal{C}$ **then return** $\pi$
**end for**

---

**Algorithm 3** Ranking via Committee Monotonicity

---

**Input:** Sampling oracle $\mathcal{D}$; committee selection subroutine COMMITTEE (Algorithm 1)
**Output:** A ranking $\pi \in \mathsf{Sym}(\mathcal{C})$
**Initialize** COMMITTEE with committee size $m$ and single-addition decomposition $(k_t = 1)_{t \leq m}$.
$\pi \leftarrow [\,]$
**for** $t = 1, \ldots, m$ **do**
  $a_t \leftarrow$ candidate added to the committee at round $t$
  Append $a_t$ to $\pi$ if it has not been added before
**end for**
Append remaining candidates to $\pi$ in arbitrary order
**return** $\pi$

---

condition, the sequence of nested committees can be naturally combined into a ranking, with candidates ordered so that each prefix coincides with a committee in the sequence and inherits its representation guarantees.

Our approach follows similar ideas, but we do not require committee monotonicity as a primitive. Instead, we begin from the ranking-level requirement and design algorithms that use committee selection as a subroutine to achieve this goal. This allows for more flexibility in how committee selection is employed.

We present two methods for constructing rankings. The first uses the committee selection algorithm as a subroutine to produce a ranking that satisfies approximate local stability with a provable constant-factor guarantee. The second is a heuristic algorithm that modifies the iterated rounding procedure in Algorithm 1 to encourage a monotone growth of selected candidates across rounds, effectively inducing a ranking by the order in which candidates are added. While this second approach does not admit strong theoretical guarantees, it has the advantage of requiring fewer user samples, and we find that it performs well empirically.

**Approach based on checkpoints.** Our first approach is based on the observation that, to ensure a ranking is approximately locally stable at every prefix, it is sufficient to guarantee that each prefix committee $W_k^\pi$ contains a smaller sub-committee $A \subseteq W_k^\pi$ that is approximately stable. Concretely, if the sub-committee is $\gamma$-approximately local stable and satisfies $|A| \geq \frac{k}{c}$ for a constant $c$, then if a user prefers $a$ to the entire prefix $W_k^\pi$, they also prefer $a$ to subset $A$, and therefore the stability of $A$ transfers to stability of $W_k^\pi$ with an additional multiplicative factor:

$$\Pr_{i \sim \mathcal{D}}[a \succ_i W_k^\pi] \leq \Pr_{i \sim \mathcal{D}}[a \succ_i A] \leq \frac{\gamma}{|A|} \leq \frac{c \cdot \gamma}{k}.$$

the classic model where the algorithm has access to each user's complete ranking. In Appendix A, we show that PSC does not extend well to the leaderboard setting in which each user's preference is accessed through limited number of pairwise comparisons.

To realize this idea, we use *checkpoint committees* to impose a blockwise structure on the ranking. We compute a sequence of approximately stable committees $A_1, \ldots, A_R$ using Algorithm 1, with geometrically increasing sizes. The ranking $\pi$ is constructed by concatenating these committees: we first set the top-ranked model as $\pi(1) = A_1$ (which is a singleton committee), and for each $r \geq 2$, append the models in $A_r$ that have not been added previously as the next block (in arbitrary order, or equivalently as ties). By construction, every prefix $W_k^\pi$ contains a checkpoint $A_{r(k)}$ whose size is within a constant factor of $k$, and the above inequality transfers its stability guarantee to $W_k^\pi$.

We formalize this construction in Algorithm 2 and state its theoretical guarantee in Theorem 4.1. The proof of Theorem 4.1 is deferred to Appendix C. Note that setting $\lambda = 2$ gives the best theoretical guarantee, but we might vary $\lambda$ in practice to control the number of ties.

**Theorem 4.1** (Guarantee of Algorithm 2). *Let $\pi$ be the output ranking of Algorithm 2 with parameter $\lambda > 1$. If the committee selection subroutine satisfies $\gamma$-approximate local stability, then the ranking $\pi$ satisfies $\frac{\lambda^2}{\lambda-1} \cdot \gamma$-approximate local stability if $\lambda$ is an integer, and $O(\frac{1}{1-\lambda}) \cdot \gamma$ if $\lambda \searrow 1$.*

**Approach based on committee monotonicity.** Our second approach is more heuristic and based on a direct adaptation of the committee selection algorithm to encourage *committee monotonicity*. Recall that Algorithm 1 selects a size-$k$ committee through a multi-round process: it first specifies a decomposition $k = k_1 + k_2 + \ldots + k_T$, and then, in each round $t \in [T]$, it selects a sub-committee of size $k_t$ for the distribution of currently unsatisfied users. The decomposition that gives rise to the theoretical guarantee in Theorem 3.1 uses a geometric schedule where $k_t \propto \alpha^t \cdot k$. While it effectively produces stable committee at a fixed size, the decomposition can change substantially when moving from size-$k$ to size-$(k + 1)$ committee. As a result, the output committees tend to violate monotonicity in general: $\hat{S}_k \not\subseteq \hat{S}_{k+1}$.

To address this issue, we consider an alternative decomposition that adds candidates one at a time. Specifically, for selecting size-$k$ committee, we set $T = k$ rounds and $k_t = 1$ for all $t \in [T]$. Under this decomposition, the committee selection process grows the committee incrementally by a single candidate in each round. This modification directly ensures committee monotonicity: constructing a size-$(k+1)$ committee is equivalent to extending the size-$k$ committee construction by one additional round, which naturally yields nested committees when randomness is shared.

Using this alternative decomposition schedule, it suffices to run a single instance of Algorithm 1 with target committee size $m$, and obtain the final ranking $\pi$ by ordering candidates according to the rounds in which they are added to the committee. We summarize this approach in Algorithm 3.

## 5. Experiments

We empirically demonstrate the performance of our committee selection algorithm and both ranking algorithms across both synthetic and semi-synthetic user distributions. We begin with fully synthetic simulations, where the goal is to empirically measure the stability of committees selected using Algorithm 1, which we find to be significantly better than the bounds given in Theorem 3.1. We then turn to a semi-synthetic experiment based on real data from LMArena (Chiang et al., 2024). We show that the currently adopted Bradley-Terry ranking indeed violates stability for some top-$k$ prefix committees, whereas our ranking is able to preserve stability for all top-$k$ prefix committees.

### 5.1. Committee Stability with Mixtures of Mallows

We construct synthetic user distributions $\mathcal{D}$ using a mixture of Mallows models (Mallows, 1957), a well-established model for user heterogeneity applied across social choice and recommender systems. We employ the standard Mallows-$\phi$ parameterization, where each mixture component consists of a central ranking $\pi$ and a dispersion parameter $\phi$, and the probability of drawing a ranking $r$ is given by $\Pr(r) \propto \phi^{d(r,\pi)}$, where $d(r,\pi)$ is the Kendall tau distance.

As a basic demonstration of the ability of Algorithm 1 to select a stable committee from a heterogeneous population, we create a mixture of $k$ Mallows models with central rankings $\{\pi_1, ..., \pi_k\}$ drawn uniformly at random from all permutations over $m = 20$ candidates, and identical dispersion parameters $\phi$ and mixture weights $\frac{1}{k}$. We then measure the approximate stability of a committee of size $k$ produced by Algorithm 1. We compare this to two baselines: *(i)* an "ideal" baseline consisting of the top-ranked candidate from each central ranking, $W_k = \{\pi_1(1), ..., \pi_k(1)\}$, and *(ii)* a "status quo" baseline consisting of the top-$k$ prefix of a Bradley-Terry ranking.

To measure stability for a given committee $W_k$, we estimate a stability approximation factor $\hat{\gamma}$ from $n$ rankings sampled from the mixture: $\hat{\gamma} = k \left( \max_{a \notin W_k} \frac{1}{n} \sum_{i=1}^n \mathbb{1} \left[ a \succ_i W_k \right] \right)$. This roughly captures the proportion of users that are unsatisfied with the committee $W_k$. A value of $\hat{\gamma} = 1$ means that no more than the required $\frac{1}{k}$ fraction of users is unsatisfied.

**Results.** Figure 1 shows the approximation factors $\hat{\gamma}$ for committees produced using Algorithm 1, as well as the two previously described baselines. Across varying $k$, we find that Algorithm 1 manages to produce committees that are stable, and with $\hat{\gamma}$ being significantly better than the bound in Theorem 3.1. In contrast, the top-$k$ prefix committee of the Bradley-Terry ranking actually violates stability for many $k$ when $\phi = 0.1$ and $\phi = 0.5$. Finally, the committee consisting of the top-ranked model in each Mallows central ranking (labeled "Mallows centers") performs well when $\phi$ is small, but as $\phi$ increases, the probability that a user's top candidate is not ranked first by one of the Mallows centers also increases. Thus, the coverage of the Mallows centers worsens as $\phi$ increases, and violates stability when $\phi = 0.9$.

### 5.2. Ranking Stability with LMArena Simulation

We next compare the stability of rankings produced by our proposed algorithms to that of a Bradley-Terry ranking, which is currently applied by the LMArena platform. We construct semi-synthetic user distributions based on the recent "arena-human-preference-140k" dataset released by LMArena (LMArena Team, 2025a; LMArena, 2025) (as our algorithms are online algorithms, we cannot run these on the static LMArena dataset directly). Candidates correspond to LLMs (of which we filter to just the top $m = 20$ by overall Bradley-Terry ranking), and users correspond to users that make pairwise comparisons between LLMs for various prompts. We construct a mixture of Mallows distribution that seeks to roughly capture the heterogeneity in rankings by prompt category. Specifically, we create a mixture of Mallows with one center for each of the 20 largest prompt categories (by category_tag), with each central ranking set to the Bradley-Terry ranking per category. The mixture weights are given by the relative number of pairwise comparisons per category. Appendix D.4 reports exact rankings and weights found for each category. As before, we conduct experiments for varying dispersions $\phi$, using the same dispersion for all categories. In essence, we model a population of users whose preferences are generally clustered around prompt categories, and higher dispersions $\phi$ correspond with higher user diversity within these clusters.

**Results.** Figure 2 shows the stability approximation factors $\hat{\gamma}$ of each of the top-$k$ committees for the Bradley-Terry ranking and the rankings produced by Algorithms 2 and 3. The Bradley-Terry ranking actually violates stability for

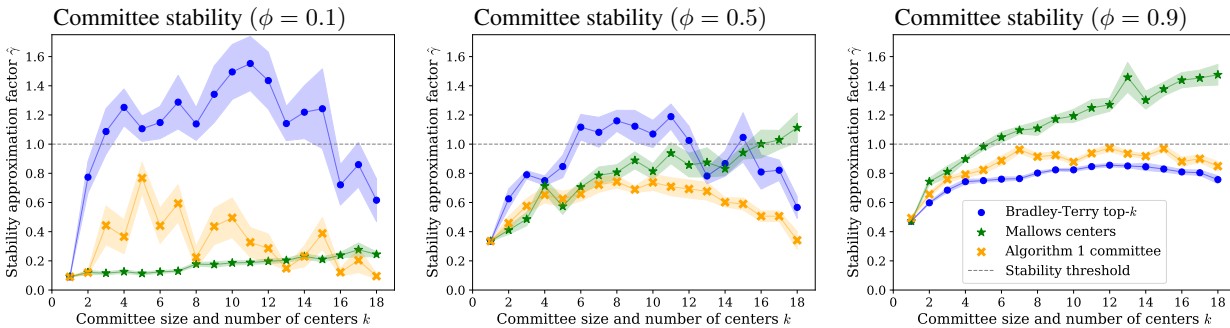

*Figure 1.* Comparison on mixtures of Mallows of the stability of committees produced by Algorithm 1 and the two described baseline methods. For each $k$, a mixture of Mallows model is created with $k$ central rankings sampled uniformly at random, and a committee of size $k$ is produced by each method. Values of $\hat{\gamma} \leq 1$ indicate that the committee produced was stable. The plotted points and error regions show the mean and standard error, respectively, after 10 repeats.

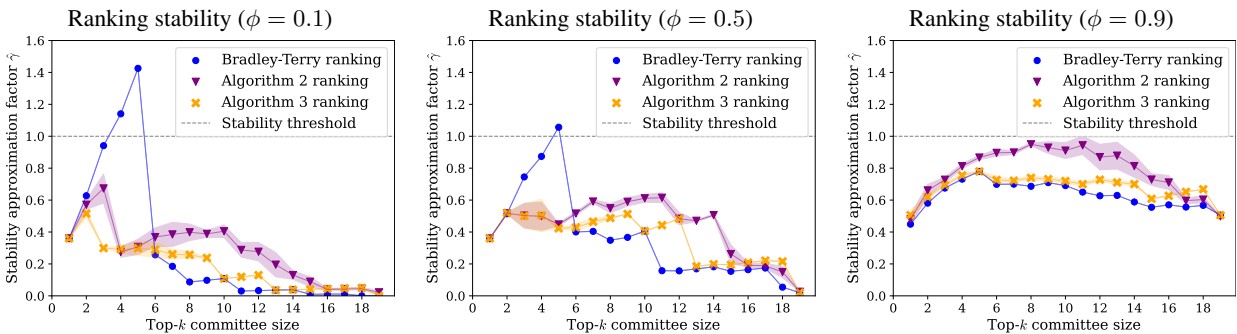

*Figure 2.* Comparison on LMArena-based simulation of the stability of Bradley-Terry rankings vs. the rankings produced by Algorithms 2 and 3. Values of $\hat{\gamma} \leq 1$ indicate that the top-$k$ prefix committee was stable for each $k$. The plotted points and error regions show the mean and standard error, respectively, after 5 repeats. A sample of exact rankings for each method is given in Appendix D.4.

top-$k$ prefix committees with $k = 4, 5$ when $\phi = 0.1$, and $k = 5$ when $\phi = 0.5$. In contrast, both Algorithms 2 and 3 maintain stability for all $k$ and all $\phi$. Intuitively, stability is "easiest" to satisfy when $\phi = 0.9$, since this corresponds to high enough user diversity that there are no large clusters of users to activate the stability constraint. While Algorithm 2 has stronger theoretical guarantees, we find that Algorithm 3 performs slightly better empirically.

## 6. Discussion

We conclude by discussing several promising directions for future research. First, our framework assumes that a user's type deterministically yields their preference ranking over models. In practice, user preferences may depend on the specific prompt as well as stochasticity in model responses. Even within a single multi-round interaction on LMArena, model pairs are often resampled for each prompt and treated as new battles, which can lead to variability in the user's preferences. Extending our framework to prompt-dependent preferences is an important direction for future work.

Second, while our proposed algorithms guarantees approximate local stability, they rely on the ability to *adaptively*

choose which model pairs are presented to users in each battle, based on both previous interactions with the same user and the history of battles contributed by other users. In many real-world settings, however, one must work with pre-collected preference datasets in which users already contribute varying numbers of comparisons. It would be interesting to study how to achieve fair representation in this offline setting.

Finally, it is natural to consider stronger notions of representation beyond local stability, such as *proportional* representation. While local stability guarantees that cohesive user coalitions of weight $O(1/k)$ are represented, proportional representation notions aim to guarantee that representation scales *proportionally* with the size of cohesive groups. For instance, Aziz et al. (2017b) also study the notion of *full local stability* which ensures any user subgroup of $O(\ell/k)$ fraction of users have no collective deviation of size $\ell$. Proportionality for Solid Coalitions (PSC) (Dummett, 1984; Aziz et al., 2025) is another prominent proportional representation notion; however, as we show in Appendix A, even access to all $k$-wise comparisons is insufficient to verify PSC for committee size of $k$. Future work could seek to understand the query complexity required to achieve such

stronger guarantees under pairwise comparison access and design efficient algorithms to do so.

## Acknowledgments

This work was partially supported by the NSF Institute for Foundations of Machine Learning under grant CCF-2505865, the National Science Foundation under grants IIS-2229881 and CCF-2145898; by the Office of Naval Research under grants N00014-24-1-2704, N00014-25-1-2153, and N00014-24-1-2159; an Amazon Research Award, an Alfred P. Sloan fellowship, a Schmidt Sciences AI2050 fellowship, an Adobe Research gift, and by grants from the Cooperative AI Foundation and the Foresight Institute.

## Impact Statement

This paper advances the evaluation of large language models by studying leaderboard design under pluralistic user preferences. By incorporating notions of fair representation from social choice theory, our work aims to make model rankings more representative for diverse user preferences and use cases. We do not anticipate significant societal impacts beyond those already associated with the use of crowdsourced human feedback for model evaluation.

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

# A. Notion for Proportional Representation

In this section, we discuss an alternative notion called Proportionality for Solid Coalitions (PSC) (Dummett, 1984; Aziz et al., 2025), which is a *proportional representation* notion that gives stronger guarantees for larger cohesive subgroups.

**Definition A.1** (Proportionality for Solid Coalitions (PSC)). A committee $W_k$ of size $k$ satisfies *Proportionality for Solid Coalitions (PSC)* for user distribution $\mathcal{D}$ if, for any candidate set $C \subseteq \mathcal{C}$, if $\Pr_{i \sim \mathcal{D}}[\forall a \in C, \forall b \notin C, a \succ_i b] > \frac{\ell}{k+1}$ for some $\ell \in \mathbb{N}$, then it holds that $|C \subseteq W_k| \geq \min\{\ell, |C|\}$.

We will show that it is challenging to verify PSC with incomplete rankings. Following Halpern et al. (2023; 2024), we consider incomplete rankings of the form of $k$-wise comparisons, where we assume that the algorithm can pick a size-$k$ subset $A \subseteq \mathcal{C}$ and ask a random user sampled from $\mathcal{D}$ to provide their ranking of $A$ in the form of a $k$-wise comparison. Note that $k$-wise comparison can be implemented using $O(k \log k)$ adaptive pairwise comparisons or $O(k^2)$ non-adaptive ones.

**Proposition A.2** (Impossibility of verifying PSC with $k$-wise comparisons). *For any committee size $k \leq m - 1$, there does not exist a deterministic algorithm that can verify whether any given size-$k$ committee satisfies PSC, using only $k$-wise comparison queries. In addition, no randomized algorithm can be correct with probability greater than $\frac{k}{k+1}$.*

*Proof of Proposition A.2.* It suffices to prove the claim in the infinite-sample limit, in which the algorithm has access to the exact marginal distribution of the user preference profile restricted to every subset $A \subseteq \mathcal{C}$ with $|A| \leq k$. Any impossibility in this setting immediately implies the same impossibility for finite samples.

We leverage the notion of $k$-*indistinguishability* introduced by Halpern et al. (2024). Two user distributions $\mathcal{D}^1, \mathcal{D}^2 \in \Delta(\mathsf{Sym}(\mathcal{C}))$ are said to be $k$-indistinguishable if

$$\mathcal{D}^1|_A = \mathcal{D}^2|_A \qquad \text{for all } A \subseteq \mathcal{C} \text{ with } |A| \leq k.$$

**Our construction.** Let $m' := k + 1$ and consider candidate subset $A = \{1, 2, \ldots, m'\} \subseteq \mathcal{C}$. By the construction of Halpern et al. (2024), there exists a family of $m'$ distributions $\{\mathcal{D}^c\}_{c \in [m']} \subseteq \Delta(\mathsf{Sym}(A))$ such that:

1. The distributions $\{\mathcal{D}^c\}_{c \in [m']}$ are $k$-indistinguishable, and

2. In each distribution $\mathcal{D}^c$, candidate $c$ is the unique candidate in $A$ whose plurality score exceeds $\frac{1}{m'}$:

$$\mathsf{plu}_{\mathcal{D}^c}(c) := \Pr_{i \sim \mathcal{D}}[c \succ_i A \setminus \{c\}] > \frac{1}{m'}; \quad \text{whereas } \mathsf{plu}_{\mathcal{D}^c}(a) < \frac{1}{m'}, \ \forall a \in A \setminus \{c\}.$$

We extend each $\mathcal{D}^c$ to a distribution $\widetilde{\mathcal{D}}^c \in \Delta(\mathsf{Sym}(\mathcal{C}))$ by appending the remaining candidates $\{m' + 1, \ldots, m\}$ to the bottom of every ranking in a fixed order. Formally, if $\sigma_i \sim \mathcal{D}^c$ is the ranking of user $i$ over $A$, then we define

$$\widetilde{\sigma}_i := \sigma_i \text{ followed by } (m' + 1) \succ_i (m' + 2) \succ_i \cdots \succ_i m,$$

and let $\widetilde{\mathcal{D}}^c$ denote the induced distribution over rankings $\widetilde{\sigma}_i$.

Next, we will show that PSC forces the inclusion of $c$. Since the additional candidates are appended below those in $A$, the plurality scores of candidates in $A$ are unchanged. In particular,

$$\mathsf{plu}_{\widetilde{\mathcal{D}}^c}(c) = \mathsf{plu}_{\mathcal{D}^c}(c) > \frac{1}{k+1}.$$

Thus, the set of users who rank $c$ first forms a solid coalition for $\{c\}$ of size exceeding the PSC threshold. Consequently, any size-$k$ committee $W_k$ that satisfies PSC under $\widetilde{\mathcal{D}}^c$ must contain $c$.

On the other hand, it is not hard to see that $k$-indistinguishability is preserved after appending these extra candidates. As a result, any algorithm that only accesses $k$-wise comparison information receives identical input (or identically distributed input in the finite-sample regime) for all distributions $\widetilde{\mathcal{D}}^c$ and therefore must output the same YES/NO for each of them.

Now Consider a committee $W \subseteq A$ of size $k$. The above argument shows that $W$ satisfies PSC under $\widetilde{\mathcal{D}}^c$ for every $c \in W$, but fails to satisfy PSC under $\widetilde{\mathcal{D}}^{c^\star}$ for $c^\star \in A \setminus W$. However, the algorithm must return the same answer for all $\widetilde{\mathcal{D}}^c$, and thus cannot be correct on all instances. This completes the proof for deterministic algorithms. The randomized bound follows from the same construction. $\square$

# B. Proof of Theorem 3.1

**Theorem 3.1** (Guarantee of Algorithm 1). *For any user distribution $\mathcal{D}$ and committee size $k$, Algorithm 1 returns a committee $\hat{S}$ of size at most $k$, such that with probability at least $1 - \delta$, $\hat{S}$ is $(16 + O(\varepsilon))$-approximately stable for $\mathcal{D}$. The algorithm has the following complexity:*

*(i) The total number of users sampled is $n = O(\mathrm{poly}(m, 1/\varepsilon, \log 1/\delta))$;*
*(ii) The maximum number of pairwise comparisons per user is $d = O(\frac{k}{\varepsilon^2} \log(\frac{m}{\varepsilon\delta}))$.*

*In particular, when $\varepsilon$ is set to a constant, Algorithm 1 achieves a constant-factor approximation using $\widetilde{O}(k)$ pairwise comparisons per user.*

We begin by proving Theorem 3.1 for the simplified setting in which the distribution $\mathcal{D}$ has finite support of size $\kappa = \mathrm{poly}(m, k, 1/\varepsilon, 1/\delta)$. We address the extension to general distributions in Appendix B.3.

Following the proof sketch in Section 3, we can apply Lemmas 3.2 and 3.3 iteratively at each round, conditioning on the success event of previous rounds and $|\mathsf{Rank}(i; S, \Delta_t) - \widehat{\mathsf{Rank}}(i; S, \Delta)| \leq \varepsilon$ for all $i \in \mathrm{supp}(\mathcal{D})$[5] and all $S \in \mathrm{supp}(\Delta_t)$. By Fact B.5, a union bound over all rounds, and the guarantee from Jiang et al. (2020) that lotteries $\Delta_t$ has support size $\mathrm{poly}(m, 1/\varepsilon)$, it suffices to implement the rank estimator using Algorithm 5 with $L = O\left(\frac{\log(T\kappa \cdot |\mathrm{supp}(\Delta)|/\delta)}{\varepsilon^2}\right) = O\left(\frac{\log(\frac{m}{\varepsilon\delta})}{\varepsilon^2}\right)$ sampled committees. This guarantees the total failure probability of the rank estimator is at most $\delta$.

Now we analyze the sample complexity of the algorithm.

- **Pairwise comparisons per user.** Algorithm 1 accesses users through the rejection sampling oracle implemented in Algorithm 4. For each user $i$, this oracle calls the rank estimator $\leq T$ times, with the sequence of $S_\tau$ and $\Delta_\tau$ in previous rounds. The total number of pairwise comparisons that $i$ being called at round $t$ needs to make is

$$O\left(\sum_{\tau=1}^{t} |S_\tau| \cdot \frac{\log(\frac{m}{\varepsilon\delta})}{\varepsilon^2}\right) = O\left(\frac{k \log(\frac{m}{\varepsilon\delta})}{\varepsilon^2}\right).$$

When $\varepsilon$ is set to be a constant, each user contributes $O(k \log(m/\delta))$ pairwise comparisons.

- **Total number of sampled users.** In each round $t$, computing the $(1 + \varepsilon)$-approximately stable lottery $\Delta_t$ using the algorithm of Cheng et al. (2020) requires $\mathrm{poly}(m, 1/\varepsilon)$ calls to the sampling oracle $\widehat{\mathcal{D}}_t^{\beta - \varepsilon}$. For the sampling oracle Algorithm 4, to make sure that the total failure probability is at bounded by $O(\delta)$, it suffices to set the number of trials to be $M = O\left(k \log(\frac{m}{\varepsilon\delta})\right)$. The next step of selecting $S_t$ from the support of $\Delta_t$ requires $N \cdot |\mathrm{supp}(\Delta_t)|$ sampled users, where $N = O\left(\log(\frac{m}{\varepsilon\delta})/\varepsilon^2\right)$ according to Lemma 3.3. Put together, the number of user sampled in each round is $O(\mathrm{poly}(m, \frac{1}{\varepsilon}, \log(\frac{1}{\delta})))$. Since there are $T = O(\log(\frac{k}{\varepsilon}))$ rounds, the total number of users sampled remains to be

$$O\left(\mathrm{poly}(m, \tfrac{1}{\varepsilon}, \log(\tfrac{1}{\delta}))\right).$$

## B.1. Proof of Lemma 3.2

**Lemma** (Restatement of Lemma 3.2). Let $S_t$ be the sub-committee selected at round $t$, and $x$ be the threshold used to measure satisfaction with the estimated rank. We have that for any alternative $a \notin S_t$,

$$\Pr_{i \sim \widehat{\mathcal{D}}_t^x} \left[a \succ_i S_t \text{ and } i \in \widehat{U}_t^x \setminus \widehat{U}_{t+1}^x\right] \leq \frac{1 + \varepsilon}{k_t \cdot (x - \varepsilon)}.$$

In other words, users who are satisfied in the $t$-th round (i.e., users in $\widehat{U}_t^x \setminus \widehat{U}_{t+1}^x$) have bounded probability of deviation.

*Proof of Lemma 3.2.* We condition throughout on the success event of Fact B.5 for lottery $\Delta_t$, under which the estimated rank $\widehat{\mathsf{Rank}}(i; S, \Delta_t)$ is an $\varepsilon$-additive approximation of the true rank $\mathsf{Rank}(i; S, \Delta_t)$ for all relevant users $i$ and all committees $S \in \mathrm{supp}(\Delta)$.

---

[5]As a result, $|\mathsf{Rank}(i; S, \Delta_t) - \widehat{\mathsf{Rank}}(i; S, \Delta_t)| \leq \varepsilon$ also holds for all $i \in \mathrm{supp}(\widehat{\mathcal{D}}_t^{\beta - \varepsilon})$ since $\widehat{\mathcal{D}}_t^{\beta - \varepsilon}$ is obtained by performing rejection sampling on $\mathcal{D}$.

We first recall a result about stable lottery from (Jiang et al., 2020, Claim 1 and Lemma 1). Suppose $\Delta$ is a $\gamma$-approximately stable lottery for $\mathcal{D}$ with committee size $k$, and $S$ is any committee in the support of $\Delta$. Then, for any threshold $x \in (0, 1)$ and any alternative $a \in \mathcal{C}$, we have

$$\Pr_{i \sim D}[a \succ_i S \text{ and } \mathsf{Rank}(i; S, \Delta) \geq x] \leq \frac{\gamma}{kx}.$$

We now apply this fact to round $t$ of the execution of Algorithm 1. Since $\Delta_t$ is an $(1 + \varepsilon)$-approximately stable lottery for $\widehat{\mathcal{D}}_t^x$, we have

$$\begin{aligned}
\Pr_{i \sim \widehat{\mathcal{D}}_t^x}\left[a \succ_i S \text{ and } i \in \widehat{U}_t^x \setminus \widehat{U}_{t+1}^x\right] &\leq \Pr_{i \sim \widehat{\mathcal{D}}_t^x}\left[a \succ_i S \text{ and } \widehat{\mathsf{Rank}}(i; S_t, \Delta_t) > x\right] \\
&\leq \Pr_{i \sim \widehat{\mathcal{D}}_t^x}[a \succ_i S \text{ and } \mathsf{Rank}(i; S_t, \Delta_t) > x - \varepsilon] \\
&\leq \frac{1 + \varepsilon}{k_t \cdot (x - \varepsilon)},
\end{aligned}$$

where the second inequality uses the $\varepsilon$-additive accuracy of $\widehat{\mathsf{Rank}}$ guaranteed by Fact B.5, and the final inequality follows from the approximate stability bound above. $\qquad\square$

## B.2. Proof of Lemma 3.3

**Lemma** (Restatement of Lemma 3.3). *For all $t \geq 1$, when the threshold is set to be $x = \beta - \varepsilon$, the probability mass of estimated unsatisfied users shrink geometrically: If $N \geq O\left(\frac{\log(mk/(\varepsilon\delta))}{\varepsilon^2}\right)$, then with probability at least $1 - \delta$,*

$$\widehat{\mu}_t := \Pr_{i \sim \mathcal{D}}[i \in \widehat{U}_t^{\beta - \varepsilon}] \leq (\beta + 2\varepsilon)^{t-1} \text{ for all } t \in [T].$$

*Proof of Lemma 3.3.* We start with proving an auxiliary lemma about selecting committees from stable lottery support based on estimated ranks, and then apply it to each round of Algorithm 1 to prove Lemma 3.3.

**Lemma B.1** (Empirical selection from a stable lottery). *For any user distribution $\mathcal{D}$ and any lottery $\Delta$, let $\mathsf{Uns}_\mathcal{D} = \Pr_{i \sim \mathcal{D}}[\widehat{\mathsf{Rank}}(i; S, \Delta) \leq \beta - \varepsilon]$ denote the unsatisfied probability under $\mathcal{D}$. Let $\widehat{\mathsf{Uns}}_\mathcal{D}$ be an unbiased estimator of $\mathsf{Uns}_\mathcal{D}$ from user samples. Specifically, given $N$ i.i.d. users $i_1, \ldots, i_N \sim \mathcal{D}$, define $\widehat{\mathsf{Uns}}_\mathcal{D}(S) = \frac{1}{N}\sum_{j=1}^N \mathbb{1}\left[\widehat{\mathsf{Rank}}(i_j; S, \Delta) \leq \beta - \varepsilon\right]$. Let $\hat{S} \in \operatorname{argmin}_{S \in \operatorname{supp}(\Delta)} \widehat{\mathsf{Uns}}_\mathcal{D}(S)$ denote the committee that minimizes the empirical unsatisfied probability.*

*If $N \geq O\left(\frac{\log(|\operatorname{supp}(\Delta)|/\delta)}{\varepsilon^2}\right)$ and $|\mathsf{Rank} - \widehat{\mathsf{Rank}}| \leq \varepsilon$ always hold, then with probability at least $1 - \delta$,*

$$\mathsf{Uns}_\mathcal{D}(\hat{S}) \leq \beta + 2\varepsilon.$$

*Proof of Lemma B.1.* By standard concentration argument and the union bound, when $N \geq O\left(\frac{\log(|\operatorname{supp}(\Delta)/\delta)}{\varepsilon^2}\right)$, we have that with probability at least $1 - \delta$,

$$\left|\widehat{\mathsf{Uns}}_\mathcal{D}(S) - \mathsf{Uns}_\mathcal{D}(S)\right| \leq \varepsilon \qquad \text{uniformly for all } S \in \operatorname{supp}(\Delta).$$

According to (Jiang et al., 2020, Lemma 1), we have the fact that for any $D, \Delta$ and threshold $\beta$, there always exists a committee $S \in \operatorname{supp}(\Delta)$ such that $\Pr_{i \sim \mathcal{D}}[\mathsf{Rank}(i; S, \Delta) \leq \beta] \leq \beta$. We denote such a committee with $S^\star$.

Now we combine the above facts to bound $\mathsf{Uns}_\mathcal{D}(\hat{S})$. We have

$$\begin{aligned}
\mathsf{Uns}_\mathcal{D}(\hat{S}) &\leq \widehat{\mathsf{Uns}}_\mathcal{D}(\hat{S}) + \varepsilon \leq \widehat{\mathsf{Uns}}_\mathcal{D}(S^\star) + \varepsilon \leq \mathsf{Uns}_\mathcal{D}(S^\star) + 2\varepsilon = \Pr_{i \sim \mathcal{D}}\left[\widehat{\mathsf{Rank}}(i; S, \Delta) \leq \beta - \varepsilon\right] + 2\varepsilon \\
&\leq \Pr_{i \sim \mathcal{D}}[\mathsf{Rank}(i; S, \Delta) \leq \beta] + 2\varepsilon \leq \beta + 2\varepsilon,
\end{aligned}$$

where the second last inequality holds since $\widehat{\mathsf{Rank}}(i; S, \Delta) \leq \beta - \varepsilon$ implies $\mathsf{Rank}(i; S, \Delta) \leq \beta$ due to the assumption $|\mathsf{Rank} - \widehat{\mathsf{Rank}}| \leq \varepsilon$. $\qquad\square$

Now we apply Lemma B.1 to prove Lemma 3.3. We do this by induction on $t \in [T]$. Consider a given round $t$ and assume that we already have $\widehat{\mu}_{t-1} \leq (\beta + 2\varepsilon)^{t-2}$. Similar to Lemma 3.2, we condition on the success event of Fact B.5 for the computed $\Delta_t$, which holds because of our choice of $N \geq O\left(\frac{\log(mk/(\varepsilon\delta))}{\varepsilon^2}\right)$ and that $|\mathrm{supp}(\Delta_t)| \leq \mathrm{poly}(m, 1/\varepsilon)$ by Jiang et al. (2020).

By applying Lemma B.1 with user distribution $\widehat{\mathcal{D}}_t^{\beta-\varepsilon}$ and the approximately lottery-committee pair $(\Delta_t, S_t)$ computed via empirical rank estimates, we have that with probability at least $1 - \delta/T$,

$$\Pr_{i \sim \widehat{\mathcal{D}}_t^{\beta-\varepsilon}}\left[i \in \widehat{U}_{t+1}^{\beta-\varepsilon}\right] = \Pr_{i \sim \widehat{\mathcal{D}}_t^{\beta-\varepsilon}}\left[\widehat{\mathsf{Rank}}(i; S_t, \Delta_t) \leq \beta - \varepsilon\right] \leq \beta + 2\varepsilon \,.$$

Then we have

$$\widehat{\mu}_{t+1} = \Pr_{i \sim \mathcal{D}}\left[i \in \widehat{U}_{t+1}^{\beta-\varepsilon}\right] = \Pr_{i \sim \mathcal{D}|\widehat{U}_t^{\beta-\varepsilon}}\left[i \in \widehat{U}_{t+1}^{\beta-\varepsilon}\right] \cdot \Pr_{i \sim \mathcal{D}}\left[i \in \widehat{U}_t^{\beta-\varepsilon}\right] \leq (\beta + 2\varepsilon) \cdot \widehat{\mu}_t \leq (\beta + 2\varepsilon)^{t-1} \,.$$

The proof is complete by taking a union bound over the failure probabilities in each round $t \in [T]$. □

### B.3. Extension to the general case

Now consider $\mathcal{D}$ that can have arbitrarily large support. We can sample $\kappa$ points from $\mathcal{D}$ and let $\tilde{\mathcal{D}}$ denote the empirical distribution over the $\kappa$ points. Then we run our proposed algorithm to select the stable committee for $\tilde{\mathcal{D}}$.

**Lemma B.2.** *When $\kappa \geq \frac{k^3 \log(m/\delta)}{\varepsilon^2}$, with probability at least $1 - \delta$, any $\gamma$-approximately stable committee for $\tilde{\mathcal{D}}$ is also $\gamma + \varepsilon$-approximately stable committee for $\mathcal{D}$.*

*Proof.* With probability at least $1 - \delta$, for any pair of $(a, S)$ where $a \in \mathcal{C}$ and $S \subseteq \mathcal{C}$ with $|S| = k$,

$$\left|\Pr_{i \sim \mathcal{D}}[a \succ_i S] - \Pr_{i \sim \tilde{\mathcal{D}}}[a \succ_i S]\right| \leq \frac{\varepsilon}{k} \,.$$

Thus, for any $\gamma$-approximately stable committee $W_k$ for $\tilde{\mathcal{D}}$, we have

$$\max_{a \notin W_k} \Pr_{i \sim \mathcal{D}}[a \succ_i W_k] \leq \max_{a \notin W_k} \Pr_{i \sim \tilde{\mathcal{D}}}[a \succ_i W_k] + \frac{\varepsilon}{k} \leq \frac{\gamma + \varepsilon}{k} \,.$$

□

**Lemma B.3** (No repetition of sampled users). *Given $\kappa \geq n^2/\delta$ points, when sampling $n$ uniformly at random from these $\kappa$ points, with probability at least $1 - \delta$, there is no repetition in $n$ samples.*

*Proof.* We denote the $n$ draws as $X_1, \ldots, X_n$, where each sample is drawn from the set of $\kappa$ points uniformly at random, and taken independently with replacement. For $1 \leq i < j \leq n$ define the event

$$E_{ij} = \{X_i = X_j\}.$$

For any fixed pair $(i, j)$ we have $\Pr(E_{ij}) = 1/\kappa$. By the union bound, the probability that there exists at least one collision satisfies

$$\Pr\left(\bigcup_{1 \leq i < j \leq n} E_{ij}\right) \leq \sum_{1 \leq i < j \leq n} \Pr(E_{ij}) = \binom{n}{2}\frac{1}{\kappa} = \frac{n(n-1)}{2\kappa} \leq \frac{n^2}{2\kappa}.$$

Since $\kappa \geq n^2/\delta$, we obtain

$$\Pr[X_1, \ldots, X_n \text{ has repetitions}] \leq \frac{n^2}{2\kappa} \leq \frac{n^2}{2 \cdot (n^2/\delta)} \leq \delta.$$

The proof is thus complete. □

---

**Algorithm 4** Rejection Sampling Oracle for Unsatisfied Users

---

**Input:** Sampling oracle $\mathcal{D}$; history of previous rounds $(S_\tau, \Delta_\tau)_{\tau < t}$; failure probability $p$; rank estimation oracle $\widehat{\mathsf{Rank}}$
**Output:** A random user $i \sim \widehat{\mathcal{D}}_t^x$, or **failure**
Number of trials $M \leftarrow \lceil k \log(1/p) \rceil$
**for** $\ell = 1$ **to** $M$ **do**
    Sample a user $i \sim \mathcal{D}$
    **if** $\widehat{\mathsf{Rank}}(i; S_\tau, \Delta_\tau) \le \beta - \varepsilon$ for all $\tau < t$ **then**
        **Return** $i$
    **end if**
**end for**
**Return failure**

---

## B.4. Additional oracles used in Algorithm 1

**Lemma B.4** (Guarantee of Algorithm 4)**.** *Suppose $\mathcal{D}$ is a user distribution, and $\widehat{\mathsf{Rank}}$ is a rank estimation oracle with error $\varepsilon$. Then, Algorithm 4 has failure probability at most $p$, and when it does not fail, the output user $i$ is from the conditional distribution $\mathcal{D}|\widehat{U}_t^x$ where $\widehat{\mathcal{D}}_t^{\beta-\varepsilon} = \widehat{U}_t^{\beta-\varepsilon}$ is the estimated unsatisfied event at round $t$. The number of sampled voters is $O(k \log(1/p))$.*

*Proof of Lemma B.4.* The fact that output users follow $\widehat{\mathcal{D}}_t^x$ is from standard fact of rejection sampling. Here, we bound the failure probability here.

At each trial, the probability of sampling a user that satisfies

$$\widehat{\mathsf{Rank}}(i; S_\tau, \Delta_\tau) \le x, \ \forall \tau \le t \quad \Longleftrightarrow \quad i \in \widehat{U}_t^x$$

is exactly $\widehat{\mu}_t = \Pr_{i \sim \mathcal{D}}[i \in \widehat{U}_t^x]$. Therefore, the failure probability after $M$ trials is at most $(1 - \widehat{\mu}_t)^M \le e^{-\widehat{\mu}_t M}$. it suffices to show that $\widehat{\mu}_t \ge \Omega(1/k)$. Note that Lemma B.1 provides an upper bound that $\widehat{\mu}_t \le (\beta + 2\varepsilon)^{t-1}$, but the actual decrease rate might be faster. To deal with this, it suffices to add a stopping condition at Algorithm 1 for early stopping as long as $\widehat{\mu}_t \le O(1/k)$. $\qquad\square$

---

**Algorithm 5** Rank Estimation Oracle

---

**Input:** Committee $S$, lottery $\Delta$, user $i$, failure probability $\delta$, error $\varepsilon$.
**Output:** An estimate of $\mathsf{Rank}(i; S, \Delta) = \Pr_{S' \sim \Delta}[S \succeq_i S']$.
Number of trials $L \leftarrow \lceil \frac{1}{\varepsilon^2} \log(1/\delta) \rceil$
Sample committees $S_1, \ldots, S_L \overset{\text{i.i.d.}}{\sim} \Delta$
**Return** $\widehat{\mathsf{Rank}}(i; S, \Delta) \triangleq \frac{1}{L} \sum_{\ell=1}^{L} \mathbb{1}\left[S \succeq_i S_\ell\right]$

---

**Fact B.5** (Guarantee of Algorithm 5)**.** *Suppose the user distribution $\mathcal{D}$ has support size at most $\kappa$, then for a fixed lottery $\Delta$, Algorithm 5 with failure probability $\frac{\delta}{\kappa \cdot |\mathrm{supp}(\Delta)|}$ guarantees that with probability at least $1 - \delta$, for any user $i \in \mathrm{supp}(\mathcal{D})$, and committee $S \in \mathrm{supp}(\Delta)$,*

$$\left| \widehat{\mathsf{Rank}}(i; S, \Delta) - \mathsf{Rank}(i; S, \Delta) \right| \le \varepsilon.$$

*The number of pairwise comparisons each user makes is $O\left(|S| \cdot \frac{\log(\kappa \cdot |\mathrm{supp}(\Delta)|/\delta)}{\varepsilon^2}\right)$.*

## C. Proof of Theorem 4.1

**Theorem 4.1** (Guarantee of Algorithm 2)**.** *Let $\pi$ be the output ranking of Algorithm 2 with parameter $\lambda > 1$. If the committee selection subroutine satisfies $\gamma$-approximate local stability, then the ranking $\pi$ satisfies $\frac{\lambda^2}{\lambda-1} \cdot \gamma$-approximate local stability if $\lambda$ is an integer, and $O(\frac{1}{1-\lambda}) \cdot \gamma$ if $\lambda \searrow 1$.*

*Proof.* Based on the choice of the sizes of checkpoint committees, the size of $A_{\leq r} = \bigcup_{s=1}^{r} A_s$ satisfies

$$|A_{\leq r}| = \left| \bigcup_{s=1}^{r} A_s \right| \leq \sum_{s=1}^{r} |A_s| \leq \sum_{s=1}^{r} \lambda^{s-1} = \frac{\lambda^r - 1}{\lambda - 1}$$

Therefore, for any prefix $W_k^\pi$ of the output ranking $\pi$, as long as

$$k \geq \frac{\lambda^r - 1}{\lambda - 1}, \hspace{3cm} \text{(Sufficient Condition)}$$

then we must have

$$W_k^\pi \supseteq A_{\leq r} \supseteq A_r.$$

It now remains to find the largest $r = r(k)$ for (Sufficient Condition) to hold. For such an $r(k)$, (Sufficient Condition) would fail for $r = r(k) + 1$, i.e.,

$$k < \frac{\lambda^{r(k)+1} - 1}{\lambda - 1}.$$

If $\lambda$ is an integer, then we have

$$\left| A_{r(k)} \right| = \lambda^{r(k)-1} \geq \frac{k(\lambda - 1)}{\lambda^2} \geq \frac{k(\lambda - 1)}{\lambda^2},$$

which implies that $W_k^\pi$, which is a superset of the $\gamma$-approximately local stable committee $A_{r(k)}$, also satisfies local stability with approximation factor $\frac{\lambda^2}{\lambda - 1} \cdot \gamma$.

When $\lambda \searrow 1$, we have

$$|A_{r(k)}| = \left\lfloor \lambda^{r(k)-1} \right\rfloor \geq \left\lfloor \frac{k(\lambda - 1)}{\lambda^2} \right\rfloor \geq \frac{k}{O(\frac{1}{\lambda - 1})},$$

which proves the second part of the theorem that $W_k^\pi$ is $O(\frac{1}{\lambda - 1}) \cdot \gamma$-approximately local stable.

$\square$

# D. Additional Experiment Details

We give additional details about the implementation of our algorithms for experiments, as well as detailed sampling procedures. All experiment code is included with the submission.

## D.1. Algorithm implementation details and default parameters

For our experimental implementation of Algorithm 1, we describe several subroutines in detail along with their default parameters used throughout the experiments. These same parameters carry forward to Algorithms 2 and 3.

**Computing an approximately stable lottery.** We describe our implementation of the following line from Algorithm 1:

$$\Delta_t \leftarrow (1 + \varepsilon)\text{-approximately stable lottery for } \widehat{\mathcal{D}}_t^{\beta - \varepsilon} \text{ of size } k_t \text{ (see Cheng et al. (2020))}$$

To implement this, we run the algorithm described by Cheng et al. (2020) in Section 2.5 using their proposed multiplicative weight update (MWU) procedure. This includes two parameters: the number of iterations $T_{\text{MWU}}$ to run the multiplicative weight update procedure, and the number of iterations $T_{\text{ORACLE}}$ to run their $\text{ORACLE}(\Delta_a, \epsilon)$ subroutine. By default in experiments, we run $T_{\text{MWU}} = 20$ iterations of the multiple weight update procedure, and $T_{\text{ORACLE}} = 30$ iterations to compute $\text{ORACLE}(\Delta_a, \epsilon)$. For each iteration of the multiplicative weight update procedure, we sample $n_{\text{MWU}} = 50$ users.

**Choosing a committee.** We describe our implementation of the following line from Algorithm 1:

$$S_t \leftarrow \text{committee in } \text{supp}(\Delta_t) \text{ that minimizes estimated unsatisfied probability}$$

In experiments, we sample $n_{\text{eval}} = 100$ users, and estimate the unsatisfied probability for each committee $S$ in $\text{supp}(\Delta_t)$.

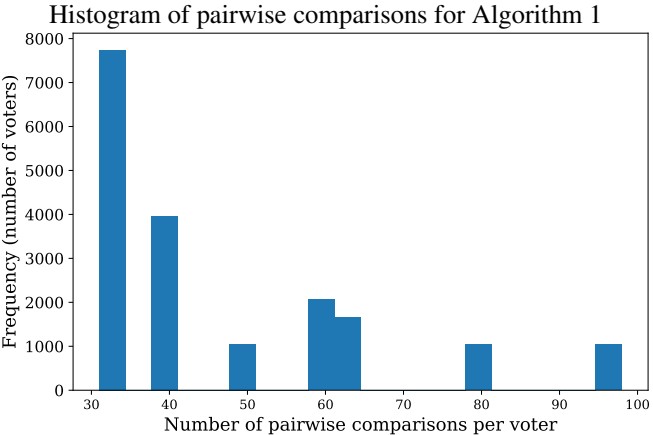

*Figure 3.* Histogram of pairwise comparisons per user for a single run of Algorithm 1 for committee size $k = 5$ and $\phi = 0.5$. The total number of users sampled was 18,540. The maximum number of pairwise comparisons for a single user was 98, and the median was 38.

### D.2. Additional details for experiments on committee stability with mixtures of Mallows (Section 5.1)

To produce the plots in Figure 1, we run the following full procedure 10 times for each $k$:

1. Sample $k$ permutations uniformly at random and set these as centers $\pi_1, ..., \pi_k$. Set mixture weights to $\frac{1}{k}$, and set all dispersions to the same given $\phi$.

2. Run Algorithm 1 with default parameters to produce a committee of size $k$. Figure 3 gives a breakdown of the samples drawn for an example run, which sampled $18,540$ users at a maximum of $98$ pairwise comparisons per user, and a median of $38$ pairwise comparisons per user.

3. Compute the Bradley-Terry ranking over a sample of $10,000$ users and all pairwise comparisons for each user, and produce a committee of size $k$ consisting of the top-$k$ ranked candidates.

4. Produce a committee consisting of $\{\pi_1(1), ..., \pi_k(1)\}$ (labeled "Mallows centers" in Figure 1).

5. Draw a sample of $n = 10,000$ users, and compute $\hat{\gamma}$ for all committees.

### D.3. Additional details for experiments on ranking stability with LMArena simulation (Section 5.2)

To produce the plots in Figure 2, we run the following procedure 5 times:

1. Run Algorithm 2 with default parameters over the data distribution to produce a ranking. Figure 4 gives a breakdown of the samples drawn for an example run, which sampled $39,080$ users at a maximum of $95$ pairwise comparisons per user, and a median of $24$ pairwise comparisons per user

2. Run Algorithm 3 with default parameters over the data distribution to produce a ranking. Figure 5 gives a breakdown of the samples drawn for an example run, which sampled $74,150$ users at a maximum of $45$ pairwise comparisons per user, and a median of $5$ pairwise comparisons per user.

3. Compute the Bradley-Terry ranking over a sample of $100,000$ users and all pairwise comparisons for each user. This is equivalent to ranking by Borda count over all sampled users, and is also equivalent to ranking by Elo-score over all possible pairwise comparisons from all sampled users. With the sample size of $100,000$ users, there was no variation in the Bradley-Terry ranking between repeats.

4. Compute the stability approximation factors $\hat{\gamma}$ for all top-$k$ prefix committees of all rankings using a draw of $n = 100,000$ users.

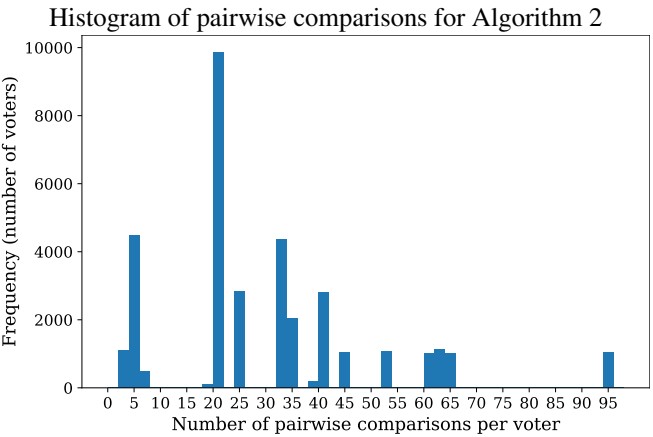

*Figure 4.* Histogram of pairwise comparisons per user for a single run of Algorithm 3 on LMArena simulation with $\phi = 0.5$. The total number of users sampled was 39,080. The maximum number of pairwise comparisons for a single user was 95, and the median was 24.

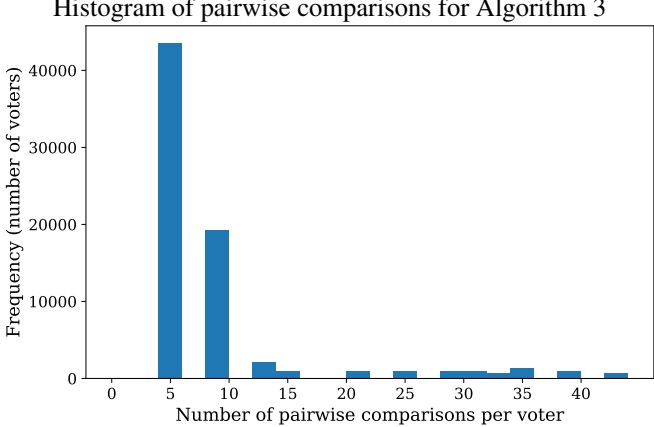

*Figure 5.* Histogram of pairwise comparisons per user for a single run of Algorithm 3 on LMArena simulation with $\phi = 0.5$. The total number of users sampled was 74,150. The maximum number of pairwise comparisons for a single user was 45, and the median was 5.

### D.4. Specific model rankings on LMArena simulation

Here we give more details the specific Bradley-Terry rankings per category from the LMArena data that form the Mallows central rankings. Table 1 shows the top two ranked LLMs per category, as well as the estimated mixture weight per category. To give a sense of the exact rankings computed in Figure 2, Table 2 shows the rankings produced by a single run of each ranking method. All full rankings can be found in the attached experiment code.

| Category index | Mixture weight | Top-ranked LLM, $\pi(1)$ | Second ranked LLM, $\pi(2)$ |
|---|---|---|---|
| 0 | 0.17 | gemini-2.5-pro-preview-03-25 | gemini-2.5-pro |
| 1 | 0.13 | gemini-2.5-pro-preview-03-25 | chatgpt-4o-latest-20250326 |
| 2 | 0.13 | chatgpt-4o-latest-20250326 | kimi-k2-0711-preview |
| 3 | 0.07 | gemini-2.5-pro | gemini-2.5-pro-preview-03-25 |
| 4 | 0.04 | gemini-2.5-pro | grok-3-preview-02-24 |
| 5 | 0.04 | chatgpt-4o-latest-20250326 | o3-2025-04-16 |
| 6 | 0.04 | deepseek-r1-0528 | llama-4-maverick-03-26-experimental |
| 7 | 0.04 | gemini-2.5-pro | deepseek-r1-0528 |
| 8 | 0.03 | gemini-2.5-pro | gemini-2.5-flash |
| 9 | 0.03 | gemini-2.5-pro | gemini-2.5-pro-preview-03-25 |
| 10 | 0.03 | llama-4-maverick-03-26-experimental | grok-4-0709 |
| 11 | 0.03 | gemini-2.5-pro | gemini-2.5-pro-preview-05-06 |
| 12 | 0.03 | chatgpt-4o-latest-20250326 | llama-4-maverick-03-26-experimental |
| 13 | 0.03 | gemini-2.5-pro | gemini-2.5-pro-preview-05-06 |
| 14 | 0.03 | gemini-2.5-pro | o4-mini-2025-04-16 |
| 15 | 0.03 | o3-2025-04-16 | llama-4-maverick-03-26-experimental |
| 16 | 0.02 | gemini-2.5-pro | llama-4-maverick-03-26-experimental |
| 17 | 0.02 | gemini-2.5-pro | grok-3-preview-02-24 |
| 18 | 0.02 | o3-2025-04-16 | gemini-2.5-pro |
| 19 | 0.02 | gemini-2.5-flash | gemini-2.5-pro |

*Table 1.* Mixture weights and top-ranked LLMs for each central ranking corresponding to a prompt category in the LMArena simulation. Each central ranking is determined by the Bradley-Terry ranking over the pairwise comparisons within a given `category_tag` from the LMArena dataset.

| $\phi = 0.1$ | |
|---|---|
| Bradley-Terry ranking | gemini-2.5-pro, chatgpt-4o-latest-20250326, grok-3-preview-02-24, o3-2025-04-16, deepseek-r1-0528, gemini-2.5-pro-preview-03-25, llama-4-maverick-03-26-experimental, gemini-2.5-flash, ... |
| Algorithm 2 ranking | gemini-2.5-pro, chatgpt-4o-latest-20250326, gemini-2.5-pro-preview-03-25, o3-2025-04-16, llama-4-maverick-03-26-experimental, o4-mini-2025-04-16, kimi-k2-0711-preview, gpt-4.1-2025-04-14, ... |
| Algorithm 3 ranking | gemini-2.5-pro, gemini-2.5-pro-preview-03-25, chatgpt-4o-latest-20250326, deepseek-r1-0528, grok-4-0709, gemini-2.5-pro-preview-05-06, o3-2025-04-16, grok-3-preview-02-24, ... |
| $\phi = 0.5$ | |
| Bradley-Terry ranking | gemini-2.5-pro, chatgpt-4o-latest-20250326, grok-3-preview-02-24, o3-2025-04-16, deepseek-r1-0528, gemini-2.5-pro-preview-03-25, llama-4-maverick-03-26-experimental, gemini-2.5-flash, ... |
| Algorithm 2 ranking | gemini-2.5-pro, chatgpt-4o-latest-20250326, gemini-2.5-flash, llama-4-maverick-03-26-experimental, gemini-2.5-pro-preview-03-25, qwen3-235b-a22b, o4-mini-2025-04-16, deepseek-r1-0528, ... |
| Algorithm 3 ranking | gemini-2.5-pro, chatgpt-4o-latest-20250326, deepseek-r1-0528, grok-3-preview-02-24, gemini-2.5-pro-preview-03-25, grok-4-0709, gemini-2.5-pro-preview-05-06, o3-2025-04-16, ... |
| $\phi = 0.9$ | |
| Bradley-Terry ranking | gemini-2.5-pro, chatgpt-4o-latest-20250326, grok-3-preview-02-24, o3-2025-04-16, deepseek-r1-0528, gemini-2.5-pro-preview-03-25, llama-4-maverick-03-26-experimental, gemini-2.5-flash, ... |
| Algorithm 2 ranking | gemini-2.5-pro, grok-3-preview-02-24, gemini-2.5-flash, llama-4-maverick-03-26-experimental, gemini-2.5-pro-preview-03-25, o4-mini-2025-04-16, chatgpt-4o-latest-20250326, qwen3-235b-a22b-no-thinking, ... |
| Algorithm 3 ranking | gemini-2.5-pro, chatgpt-4o-latest-20250326, gemini-2.5-pro-preview-03-25, o3-2025-04-16, deepseek-r1-0528, grok-3-preview-02-24, llama-4-maverick-03-26-experimental, grok-4-0709, ... |

*Table 2.* Example rankings from a single run of each ranking in Figure 2. For ease of visualization, we show the top 8 ranked candidates per ranking here. This does not account for variation across repeats, but gives a single ranking sample as an illustration of the tendencies of each ranking method.

