# OpenReview forum: "Pluralistic Leaderboards"
_ICML.cc/2026/Conference — ICML 2026 regular_

### Official Review · Reviewer_t12x · 2026-03-08

**Soundness:** 3
**Presentation:** 3
**Significance:** 2
**Originality:** 3
**Overall Recommendation:** 4
**Confidence:** 2

**Summary:**

The paper designs a pluralistic leaderboard framework that is compatible with heterogenous preferences.

**Compliance With Llm Reviewing Policy:**

Affirmed.

**Key Questions For Authors:**

1. For the committee selection algorithm, do you assume the underlying distribution of user population is known or unknown? How did you estimate it?
2. When randomly select permutations form all candidates, how many permutations do you select and how does it varies when the number of candidates is getting larger? Will it add any computation burden as the size increasing?

**Limitations:**

Yes

**Strengths And Weaknesses:**

Pros: The paper is well-structured, clearly states the question, and explain all concepts in a simple way, which makes readers easy understand.
Cons: It will be better if the author can further provide a diagram/flowchart to explain the designed pluralistic leaderboards framework.

---

> ### Author Rebuttal · Authors · 2026-03-30
>
> > For the committee selection algorithm, do you assume the underlying distribution of user population is known or unknown? How did you estimate it?
>
> The underlying distribution over user preferences is unknown. We assume only sampling access to users, along with query access to each sampled user’s preferences via pairwise comparisons. Our algorithms operate directly under sampling access, without explicitly estimating the user distribution first. Indeed, the space of possible rankings has size $m!$, so explicitly estimating the distribution is potentially harder than the task of constructing a stable committee for that distribution.
>
> > When randomly select permutations form all candidates, how many permutations do you select and how does it varies when the number of candidates is getting larger? Will it add any computation burden as the size increasing?
>
> As we show in Theorem 3.1 part (i), the number of users (where each user’s preference can be represented by a permutation) is polynomial in the number of candidates. The computational complexity is also polynomial in the number of candidates.

---

> > ### Author Rebuttal · Reviewer_t12x · 2026-04-01
> >
> > Thanks for your response.

---

### Official Review · Reviewer_BvkA · 2026-03-13

**Soundness:** 3
**Presentation:** 3
**Significance:** 4
**Originality:** 4
**Overall Recommendation:** 5
**Confidence:** 4

**Summary:**

Article deals with evaluation of LLMs through pluralistic leaderboards that take pairwise comparisons as input and produce an outcome that fairly represents heterogeneous preferences.

	Motivation for this study is that the current leading approach for evaluating LLMs relied on benchmarks, which measures performance based on multiple tasks and aggregates the corresponding scores. This approach has consistently provided a standardized and reproducible basis for comparing models but is becoming increasingly strained by rapid model iteration, benchmark saturation, and sensitivity to task selection. It also struggles to capture qualitative dimensions of model behavior such as helpfulness.

	Article notes that some of these shortcomings are addressed by alternative evaluation approaches like LMArena, which produces a model score through human-input and fitting the votes through a Bradley-Terry (BT) model. This approach, however, attracts criticism because of the implicit assumption that all pairwise comparisons originate from a generic (or "average) person, "a view that is fundamentally at odds with heterogeneous preferences."

	Article notes that people come to LLMs with different goals (e.g. safety vs creativity, conciseness vs thoroughness, formality vs warmth) that represents inherently different evaluation metrics. Social choice, as the article notes, provides a principled framework for aggregation.

	Article next introduces his or her notion of pluralistic leaderboards for fair representation, which author argues requires local stability to be required to produce rankings, treating the top-$k$ prefix of a ranking as size $k$ committee and impose local stability at every cutoff $k$. The goal is, therefore, to design ranking mechanism that converts pairwise comparisons into an (approximately) local stable ranking. The algorithm, according to the author, builds upon algorithms by Jiang et al, where users each contribute only a small amount of feedback, rather full rankings. Author then proposes an algorithm based on finite-sample estimation errors control, that produces approximately local stable committees, with a constant approximation factor. A reduction to the algorithm applied so that algorithm is used as a subroutine to construct approximate stable rankings, incurring only a constant multiplicative factor in the approximation ration.

	The algorithm's notable characteristics include
	- the Leaderboard Mechanism, where rankings are produced through pairwise "battles," pairwise comparisons between two models and a ranking mechanism that samples across battle population D and query each sampled user for preferences
	- Bradley-Terry Ranking by LM Arena, applying active sampling for selecting model pairs for ranking of models according to scores
	- Local Stability ensuring no outside model is collectively preferred by sufficiently large subgroup of users noting local ranking stability for user distribution for every $k$<$m$
	- Committee selection - where a sampling-based Iterated Rounding Algorithm was applied to select a single committee of a given size that satisfies the approximate stability while accounting for fair representation.

	Article then proceeds to provide a proof of stability by comparing query quantity ranks estimated through user  $i$ to compare $S$ against a finite number of committees. The algorithm selects a number of sub-committees at each round and calculates a satisfaction level assumed at each rank which is in turn decomposed by user population into T+1 sub-groups. The t-th subgroup is the users that become satisfied at round t, and iterating for the remaining users.

	Finally, parameters are optimized to minimize approximation factor for committee selection and the rankings are outputted by selecting top rankings from locally stable committees. The article makes use of two approaches
	1. Ranking via Geometric Checkpoints - Using committee selection as a subroutine to produce a ranking that satisfies the approximate local stability with a provable constant-factor guarantee.
	2. Ranking via Committee Monotonicity - using a heuristic algorithm that modifies the iterated rounding procedure in Algorithm 1 to encourage monotonic growth of selected candidates across rounds, effectively inducing a ranking by the order in which candidates are added.
	Author notes that the second approach does not yield strong theoretical guarantees, it has the advantage of requiring fewer user samples and it was found to perform empirically well.

	Experiments performed include:
	- Committee Stability with Mixtures of Mallows - Synthetic user distributions D are constructed using a mixture of Mallow models, a well-established model and used to measure stability for given committee based on dispersion $\psi$. Author's experimentation show that the top-$k$ BT ranking actually violate stability for the parameters experimented by author. In contrast, the top-$k$ ranked models for allows central ranking performs well in comparison but falls off when $\psi$ increases, finally violating stability when $\psi=0.9$.
	- Ranking Stability with LMArena Simulation - next, stability rankings are compared using that of a Bradley-Terry ranking, which is currently applied to LMArena platform. Pairwise LLM-prompt comparisons were made and a mixture of Mallows distribution vs BT ranking category were made. In essence, a population of users whose preferences are generally clustered around prompt categories and higher dispersions $\psi$ correspond with higher user diversity within clusters.	Results show that stability approximation factors of each top-$k$ committees for Bradley-Terry ranking and rankings produced by Ranking via Geometric Checkpoints and Ranking via Committee Monotonicity actually violates stability while both Ranking algorithms maintain stability for all $k$ and $\psi$. Stability is easiest to satisfy when  $\psi=0.9$ since this corresponds to high enough user diversity that there are no large clusters of users to activate the stability constraint. Ranking via Geometric Checkpoints is shown to be slightly inferior to Ranking via Committee Monotonicity.

	A discussion on future directions include deterministically yielding preference ranking over models. Author notes that in practice, user preferences may depend on the specific prompt as well as stochasticity in model responses so a natural extension of the proposed algorithm would be to extend the framework to prompt-dependent preferences.
	Author also notes that although algorithm guarantees approximate local stability, the reliance on the ability to adaptively choose which model pairs are presented to users limits utility in real-world settings, where one must work with pre-compiled preference datasets, motivates study in how to achieve fair ranking representation in offline settings.
	Finally, author proposes study on 'stronger' representations beyond local stability such as $proportional representation$, full local stability, proportionality for solid coalitions (PSC). Author notes even access to all $k$-wise comparisons is insufficient to verify PSC for committee size $k$ and proposes that future work could seek to understand query complexity to achieve stronger guarantees under pairwise comparison access and design efficient algorithms to do so.

**Compliance With Llm Reviewing Policy:**

Affirmed.

**Key Questions For Authors:**

Would it be possible to approach this problem more iteratively?

**Limitations:**

yes

**Strengths And Weaknesses:**

Soundness: Reasoning for each algorithm design decision is technically sound, but design methods appear to be archaic with metrics' origins seemingly inserted. I know these topics tend to be relatively difficult to quantify and doing so would require lots of extra steps to produce.
Presentation: This motivation for each algorithm is well articulated. I would like to see more discussion as to why each design decision works
Significance: Ranking LLMs is a difficult task and I applaud the author for tackling such a complex subject while approaching the subject as rigorously as he could
Originality: Very original work

---

> ### Author Rebuttal · Authors · 2026-03-30
>
> > Reasoning for each algorithm design decision is technically sound, but design methods appear to be archaic with metrics' origins seemingly inserted. I know these topics tend to be relatively difficult to quantify and doing so would require lots of extra steps to produce.
>
> We want to clarify that the use of *classical* design methods such as local stability is intentional, since the core challenge of aggregating heterogeneous preferences into a single ranking is fundamentally a social choice problem. We aim to build on these well-established principles to ensure that our framework is grounded. Our technical contribution is in adapting these classical tools to the LLM leaderboard setting with only pairwise comparisons, which introduces new challenges around sample efficiency.
>
>
> > Would it be possible to approach this problem more iteratively?
>
> The committee selection algorithm relies on iteratively reducing the fraction of unsatisfied users. Our approaches for extending committee selection to rankings are also iterative, as described in Section 4.

---

> > ### Author Rebuttal · Reviewer_BvkA · 2026-04-06
> >
> > To reframe my question, why was pairwise comparisons used ..

---

> > > ### Author Response · Authors · 2026-04-07
> > >
> > > > why was pairwise comparisons used ..
> > >
> > > This is because, on current LLM leaderboards such as LMArena, pairwise comparisons are essentially the only reliable signal we have about human preferences. It is too difficult for users to provide well-calibrated numerical scores, and it is also impractical to ask them to provide full rankings on too many models at once. As a result, existing leaderboards rely on pairwise comparisons as the primary way to collect preference data.

---

### Official Review · Reviewer_Nkif · 2026-03-19

**Soundness:** 3
**Presentation:** 3
**Significance:** 3
**Originality:** 3
**Overall Recommendation:** 4
**Confidence:** 2

**Summary:**

This paper starts by arguing that standard LLM leaderboards based on a single Bradley–Terry (BT) score are conceptually mismatched to heterogeneous user preferences, because collapsing all pairwise comparisons into one latent quality can systematically misrepresent subgroups with different goals. To address this issue, the authors introduce pluralistic leaderboards, which borrows the idea of "social-choice notion of local stability" and extending it from committees to ranking prefixes. What's more, the authors develop an iterated-rounding-style committee selection algorithm and build two ranking constructions on top of it: one with provable approximation guarantees via geometric checkpoints, and one heuristic based on committee monotonicity. Empirically, the authors show that Bradley–Terry top-$k$ prefixes can violate stability, whereas their methods maintain stability much more reliably.

**Compliance With Llm Reviewing Policy:**

Affirmed.

**Key Questions For Authors:**

Please refer to my summary of weaknesses.

**Limitations:**

yes

**Strengths And Weaknesses:**

This paper has the following advantages that I appreciate very much:

+ In my opinion, using one latent BT style score for each LLM can indeed cause problems in under representing the heterogenous user preferences, so the paper has very strong and compelling motivations.

+ The idea of adapting the social-choice notion of local stability and extending it from committees to ranking prefixes to support the ranking of different LLM based on user's prompt and choices seems novel.

Meanwhile, the method also has the following weaknesses:

- Although the paper frames the problem broadly as leaderboard design under heterogeneous preferences, the experimental validation is still fairly narrow: the semi-synthetic study is based on a Mallows-mixture simulation built from LMArena categories rather than a fully realistic online deployment or real user-level longitudinal setting. I understand that this is difficult but I'm wondering whether small-scale experiments would be possible.

- I'm curious whether the proposed method is robust to noise in pair-wise comparison. For example, would the model work if two responses are similar where the choice from certain users are more random compared to the case where two responses are drastically different in quality.

---

> ### Author Rebuttal · Authors · 2026-03-30
>
> > Although the paper frames the problem broadly as leaderboard design under heterogeneous preferences, the experimental validation is still fairly narrow: the semi-synthetic study is based on a Mallows-mixture simulation built from LMArena categories rather than a fully realistic online deployment or real user-level longitudinal setting. I understand that this is difficult but I'm wondering whether small-scale experiments would be possible.
>
> Our choice of a semi-synthetic setup is driven by two constraints:
> 1. Existing public LMArena datasets either lack user identity information, or contain user information but exhibit large variation in the number of comparisons per user, where the majority of users contribute very few comparisons. This makes it difficult to model heterogeneity at the user level or to enable adaptive querying.
> 2. Our algorithm requires adaptively choosing comparison pairs based on each user’s previous responses, so we cannot run it on static, precollected datasets.
>
> Therefore, even a small-scale experiment would require deploying an interactive system that supports adaptive querying and interfaces with many LLMs. This is not impossible, but it is clearly a significant undertaking.
>
>
> > I'm curious whether the proposed method is robust to noise in pair-wise comparison. For example, would the model work if two responses are similar where the choice from certain users are more random compared to the case where two responses are drastically different in quality.
>
> The reviewer is suggesting a **utilitarian** perspective, where each user implicitly assigns a quality (utility) score to each model-generated response, and pairwise comparisons are generated from a random utility model (e.g., Bradley--Terry). As a result, when two responses have similar utility for a given user, comparisons are more random, and when they are far apart, comparisons are more deterministic.
>
> Our framework instead takes a **non-utilitarian** perspective: we do not assume such a latent quality score or any specific random utility model. Instead, we take the observed preferences as given and define guarantees directly with respect to the distribution of preferences (rankings). As a result, our theoretical guarantees are agnostic to the noise level in pairwise comparisons.
>
> Prior work has investigated the utilitarian perspective in the context of single-model alignment. In particular, Gölz et al. [2025] study how performance depends on the noise level. From that perspective, analyzing stability under such models would be an interesting direction for future work, but it is orthogonal to our focus.
>
> **References**
>
> Gölz, P., Haghtalab, N., & Yang, K., 2025. Distortion of AI Alignment: Does Preference Optimization Optimize for Preferences? In NeurIPS'25.

---

> > ### Author Rebuttal · Reviewer_Nkif · 2026-04-02
> >
> > Thank you for your detailed responses to address my concerns.

---

### Decision · Program_Chairs · 2026-04-30

**Decision:**

Accept (regular)

**Comment:**

This paper addresses a timely problem in LLM evaluation: standard leaderboards aggregate pairwise user comparisons via a Bradley-Terry model into a single global ranking, which is fundamentally incompatible with heterogeneous user preferences. The authors propose pluralistic leaderboards grounded in social choice theory, adapting the notion of local stability from committee selection to ranking prefixes — requiring that no model outside the top-k is collectively preferred to the top-k set by more than an O(1/k) fraction of users. The paper designs an iterated-rounding-based algorithm and two ranking constructions (geometric checkpoints with provable guarantees and committee monotonicity as a practical heuristic), requiring only Õ(k) pairwise comparisons per user. Using LMArena data with Mallows-mixture user models, the authors demonstrate that BT ranking violates local stability in practice, whereas their method maintains stability across diverse user-diversity regimes.

All three reviewers were positive. They recognized the strength and timeliness of the motivation, praised the originality of adapting local stability to the LLM leaderboard setting, and found the algorithmic design principled and the presentation clear. Concerns were minor: the semi-synthetic nature of the experimental setup, questions about noise robustness under a utilitarian model, and computational complexity. The authors addressed all concerns effectively in the rebuttal — clarifying the non-utilitarian framing, explaining why adaptive querying precludes static datasets, and confirming polynomial computational complexity — both are fully resolved.

I therefore recommand acceptance.